# The molecular basis of regulation of bacterial capsule assembly by Wzc

Yun Yang[1,2,3,8], Jiwei Liu[1,2,8], Bradley R. Clarke [4], Laura Seidel[4], Jani R. Bolla [5,6], Philip N. Ward [1,2,3], Peijun Zhang [2,7], Carol V. Robinson [5,6], Chris Whitfield [4✉] & James H. Naismith [1,2,3✉]

Bacterial extracellular polysaccharides (EPSs) play critical roles in virulence. Many bacteria assemble EPSs via a multi-protein "Wzx-Wzy" system, involving glycan polymerization at the outer face of the cytoplasmic/inner membrane. Gram-negative species couple polymerization with translocation across the periplasm and outer membrane and the master regulator of the system is the tyrosine autokinase, Wzc. This near atomic cryo-EM structure of dephosphorylated Wzc from *E. coli* shows an octameric assembly with a large central cavity formed by transmembrane helices. The tyrosine autokinase domain forms the cytoplasm region, while the periplasmic region contains small folded motifs and helical bundles. The helical bundles are essential for function, most likely through interaction with the outer membrane translocon, Wza. Autophosphorylation of the tyrosine-rich C-terminus of Wzc results in disassembly of the octamer into multiply phosphorylated monomers. We propose that the cycling between phosphorylated monomer and dephosphorylated octamer regulates glycan polymerization and translocation.

[1] Rosalind Franklin Institute, Harwell Campus, Harwell, UK. [2] Division of Structural Biology, The University of Oxford, Oxford, UK. [3] The Research Complex at Harwell, Harwell Campus, Harwell, UK. [4] Department of Molecular and Cellular Biology, The University of Guelph, Guelph, ON, Canada. [5] Physical and Theoretical Chemistry Laboratory, Department of Chemistry, South Parks Road, The University of Oxford, Oxford, UK. [6] The Kavli Institute for Nanoscience Discovery, Oxford, UK. [7] Electron Bio-Imaging Centre, Diamond Light Source, Harwell Science and Innovation Campus, Harwell, UK. [8] These authors contributed equally: Yun Yang and Jiwei Liu. ✉email: cwhitfie@uoguelph.ca; naismith@strubi.ox.ac.uk

High-molecular-weight extracellular polysaccharides (EPSs) play prominent roles in interactions between bacteria (pathogens, commensals, and symbionts) and their hosts (humans, livestock, and plants). Some EPSs are also important bioproducts in foods and other commercial applications. Although the structures of bacterial surface-associated and secreted polysaccharides are remarkably diverse, the majority are produced by a conserved and widely distributed assembly strategy (the Wzx-Wzy system) (reviewed in ref. [1]). The process begins with the synthesis of polyprenol diphosphate-linked oligosaccharide repeat units at the cytoplasmic face of the inner membrane (IM). The lipid-linked repeat units are then flipped across the IM by the Wzx flippase, where they act as substrates for the Wzy polymerase, an integral membrane protein with a catalytic site located at the external face of the IM. The subsequent assembly stages differ, depending on the bacterial species and the glycoconjugate, but all involve a polysaccharide co-polymerase (PCP) protein[2] that is important for establishing the size distribution of the polymeric product. This is vital for the biophysical properties and biological functions of the polymers. In the production of capsular polysaccharides (CPS) or secreted EPSs in Gram-negative bacteria (Fig. 1a), Wzc (PCP-2a) proteins are thought to be the master regulator for both polymerization and translocation but the mechanism of regulation remains one of the most important questions in this field[1]. Wzc possesses a C-terminal cytosolic protein tyrosine kinase (PTK) belonging to the BY-kinase family[3], which contains Walker A and B motifs, as well as a tyrosine-rich tail presenting several residues for phosphorylation (reviewed in ref. [2]) (Fig. 1a). In contrast, the biosynthesis of lipopolysaccharide (LPS)-linked O-antigen polysaccharides by Wzx-Wzy pathways involves Wzz (PCP-1) homologs that lack a kinase domain and possess a shorter divergent periplasmic sequence compared to Wzc[2,4]. This reflects some functional differences; Wzz is important for regulating O-antigen polymer chain length but plays no apparent role in the translocation of the final products[4]. Structural data are available for Wzz[5–7] but not for full-length Wzc.

In *Escherichia coli* EPS assembly prototypes, the Wzc autokinase is paired with a cognate soluble protein tyrosine phosphatase (PTP)[1,2]. The structure of one PTP, Wzb, has been determined[8,9]. The corresponding genes are found in the EPS biosynthesis genetic locus but many isolates also possess unlinked genes encoding an additional Wzc and Wzb homolog pair (designated Etk and Etp)[10]. Autophosphorylation of the C-terminal tyrosine-rich tail of Wzc is essential for polymerization of the EPS product[11–13]. Paradoxically, the removal of phosphates from Wzc by Wzb is also essential[11–13]. The requirement for both kinase and phosphatase activity has led to the hypothesis that the cycling between different Wzc phosphorylation states is critical for polymer production, rather than any one state[13–15]. This hypothesis holds in Gram-positive bacteria, which employ bipartite PCP-2b proteins resembling Wzc[1]. Crystal structures of isolated kinase domains from *E. coli* Wzc[16] and *S. aureus* CapAB chimera[17] reveal octameric rings in which the non-phosphorylated C-terminal tail from one monomer is positioned in the active site of its neighbour. In contrast, Etk is not oligomeric in the presumed phosphorylated state[18]. The lack of a structure of the full-length protein where the kinase domains are in context is a major gap in functional understanding.

In Gram-negative bacteria, EPS must traverse the periplasm and outer membrane (OM) from the site of synthesis and this requires the octameric Wza translocon[19,20] (Fig. 1a). Genetic data implicate specific Wza:Wzc interactions in EPS production and translocation[21] and oligomeric Wza:Wzc heterocomplexes were visualized in negatively-stained EM images[22]. The working hypothesis is that the periplasmic domain of Wzc engages with the periplasmic region of the Wza to form the structural machinery for translocation and EPS exits through the lumen of the translocon[20]. The absence of an OM in Gram-positive bacteria precludes this Wza interaction and CapA (for example) possesses a substantially reduced periplasmic domain.

Our understanding of the important physiological process of EPS biosynthesis and translocation is hindered by a lack of molecular insight into the master regulator, Wzc. Here we report the near-atomic structures of non-phosphorylated Wzc octamers from *E. coli*, in the apo- and ADP-bound states. We establish that the pattern of tyrosine phosphorylation controls the oligomerization state and demonstrate that periplasmic helical bundles are essential for function. We advance a mechanistic model for the EPS production system found across the bacterial kingdom, which is important for biotechnological applications and may offer options for targeted therapeutics.

## Results

**Non-phosphorylated full-length K540M Wzc is an octameric protein.** Native tyrosine-phosphorylated Wzc was purified but the protein eluted as a broad peak in gel filtration and negatively-stained EM images showed heterogeneity in the samples (Supplementary Fig. 1a–c). Native protein mass spectrometry detected predominantly monomeric species and abundant phosphorylation adducts possessing 3, 4 or 5 phosphate groups (Fig. 1b). Further phosphoproteomic analysis suggested that the penta-phosphorylated species (5 P) was modified on tyrosines 708, 713, 715, 717 and 718. The strongest peak in tetra-phosphorylated (4 P) protein was modified on residues 708, 713, 715 and 717, but a species with phosphotyrosines at 708, 713, 717 and 718 were also detected. The triple-phosphorylated (3 P) was more heterogeneous (Supplementary Fig. 2a, b).

To test the possibility that phosphorylation led to heterogeneity and aggregation of purified Wzc, a sample was dephosphorylated by treatment with phosphatase Wzb and then reapplied to the size exclusion column but it remained like the native protein (Supplementary Fig. 1d–f). Previous work showed that mutation of K540 in the Walker box abrogated Wzc phosphorylation[13,17]. In contrast to phosphorylated and enzymatically-dephosphorylated protein, purified non-phosphorylated Wzc[K540M] was homogeneous, oligomeric (based on molecular mass in gel filtration) and yielded discrete octameric particles in negatively-stained EM images (Supplementary Fig. 1h–j). We re-examined the Wzb treated sample and collected 1787 cryo-EM images which were analysed by Topaz[23] to identify 356,511 particles (Supplementary Fig. 1g). Through rounds of reference-free 2D classifications, 421 particles converged into clear views of an octameric Wzc (Supplementary Fig. 1g). In the raw micrographs, these particles (Supplementary Fig. 1g) resembled octamers of Wzc[K540M]. However, the small octameric population (~0.1%) in the Wzb-treated sample made structure determination impractical, so we proceeded with Wzc[K540M].

A cryo-EM structure of Wzc[K540M] at 2.85 Å resolution was determined with C1 symmetry (Supplementary Table 1, Supplementary Fig. 3a–e). Although the data were processed in C1, much of the structure displayed eightfold symmetry and the application of C8 symmetry improved the resolution to 2.30 Å (Supplementary Fig. 3f, g). However, C8 symmetry breaks down for part of the periplasmic region, so local C1 refinement was used to complete the structure (Supplementary Fig. 3a, h, i). Wzc[K540M] in complex with ADP was determined to 2.6 Å with C8 symmetry (Supplementary Fig. 4a–d) and is essentially identical to the apo-structure, with the exception of additional density for ADP and Mg$^{2+}$ (Supplementary Fig. 4e). Below, we focus on the higher resolution apo-structure.

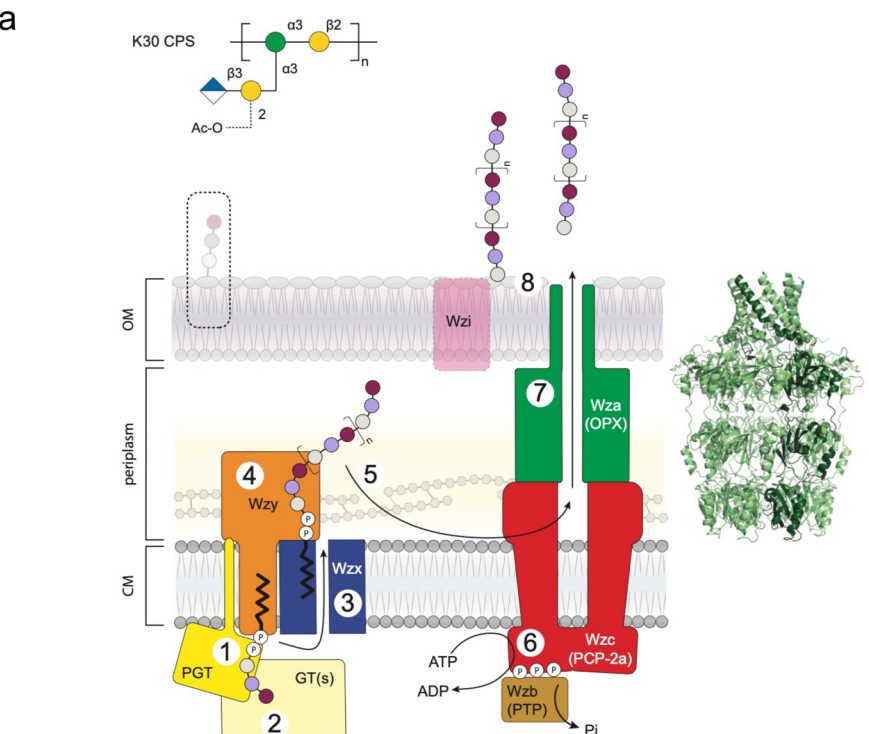

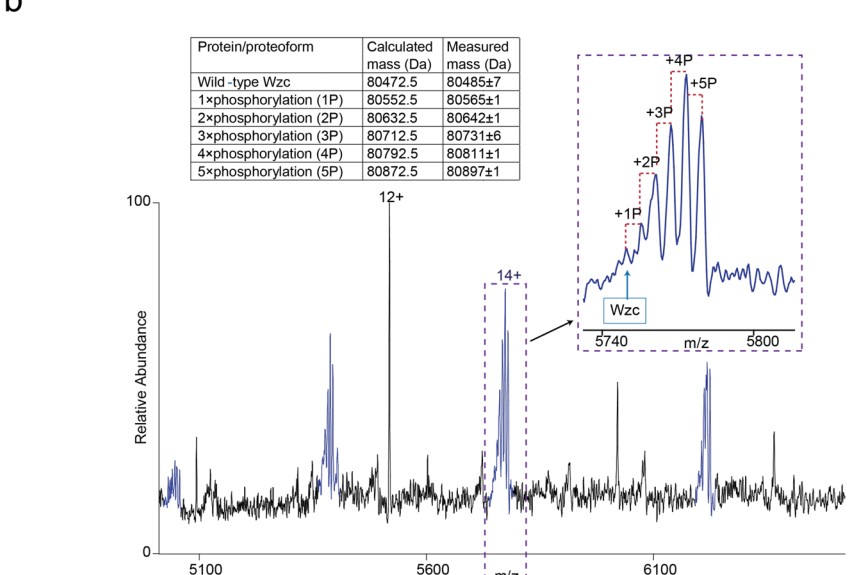

**Fig. 1 Capsule assembly and Wzc. a** Undecaprenyl diphosphate-linked oligosaccharide repeat units are synthesized by a phosphoglycosyltransferase (PGT; (1)) and serotype-specific glycosyltransferases (GT(s); (2)). These building blocks are flipped across the membrane by Wzx (3) and polymerized by Wzy (4) in a reaction regulated by the PCP-2a family autokinase, Wzc. Wzc cycles between phosphorylated and dephosphorylated (catalysed by Wzb) states for its function (6). The polymer is translocated across the outer membrane by Wza (7); the closed octameric structure of Wza[19] is shown on the right. Wza is also regulated by Wzc. In the prototype and some other species, Wzi supports the organization of translocated polymer into the surface-associated capsule structure (8) but this protein is absent from systems that produce secreted EPSs. In Gram-positive, bacteria step (7) is absent and nascent polymer is instead attached to peptidoglycan in the cell wall. In the absence of Wzc or Wza, short oligosaccharides representing one to a few repeat units are incorporated into LPS molecules in an off-pathway reaction (hatched box). The process has recently been reviewed[1,4]. The figure has been adapted from[1]. **b** Purified, native phosphorylated Wzc appears only as a monomer (peak series highlighted in blue) in native mass spectrometry. The 14[+] charge state shows multiple phosphorylation (denoted P) states of Wzc with four sites being the most prevalent (inset). The theoretical and measured masses for all the observed species are shown in the table. Supplementary Fig. 2a identifies phosphopeptides. The peak series with a measured mass of 66,212 ± 2 Da was a contaminant protein.

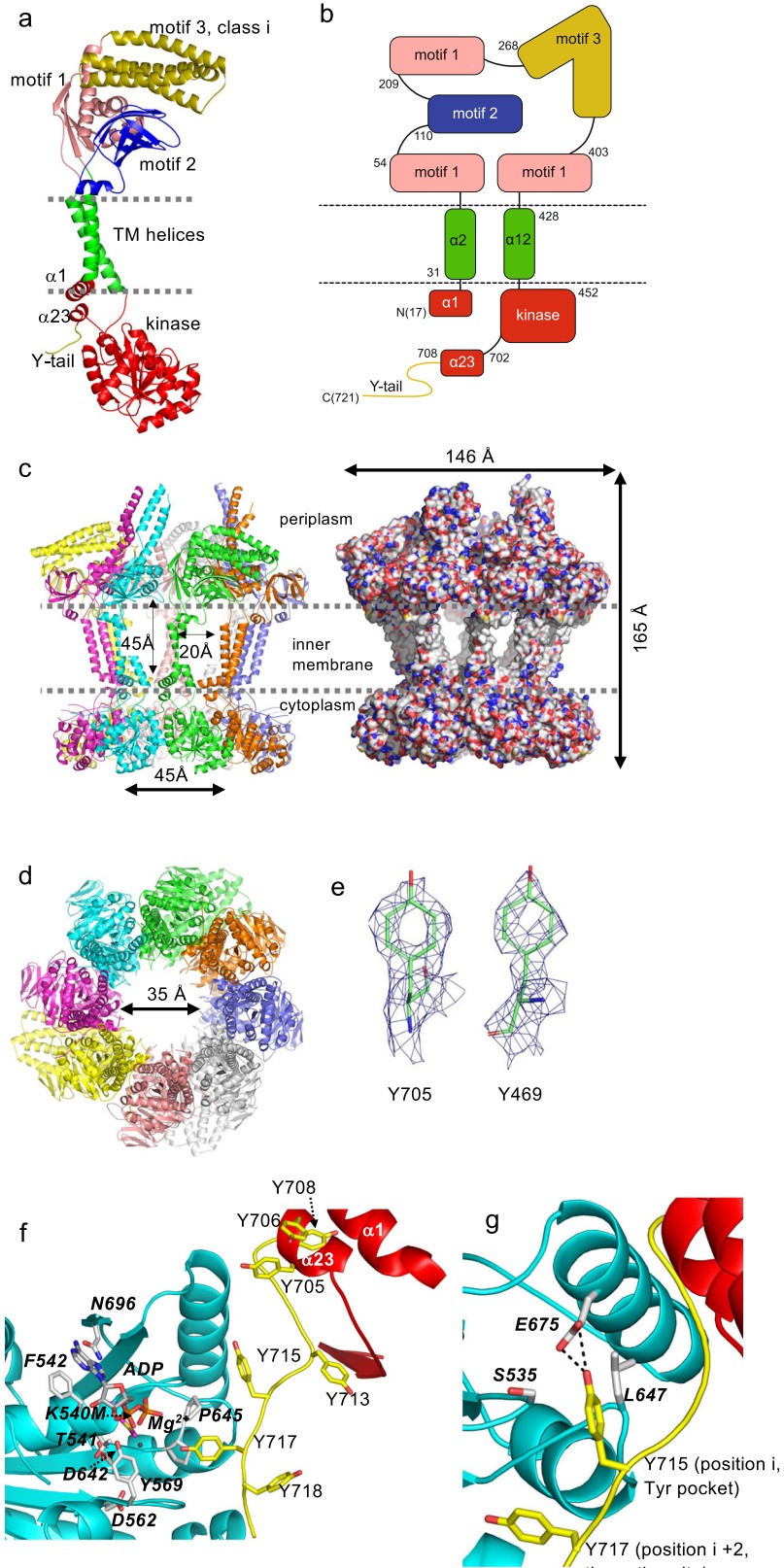

For ease of discussion, each protomer can be decomposed into three portions: the periplasmic region (residues 51–427), the transmembrane region (31–50, 428–447) and the cytoplasmic region (17–30, 448–721) (Fig. 2a, b). The Wzc$^{K540M}$ octamer is ~165 Å high (cytosol to periplasm) and 146 Å wide and possesses a large central cavity with side portals that open to the membrane bilayer (Fig. 2c, d). The high quality of the experimental map (Fig. 2e and Supplementary Fig. 3j–m) allows accurate assignment of the sequence to structural elements. The cytoplasmic region comprises a short N-terminal helix (α1, Ile 17-Arg 30), the

**Fig. 2 The structure of Wzc$^{K540M}$ determined by single-particle cryo-EM. a** The monomer of Wzc$^{K540M}$ contains a cytoplasmic region (red), tyrosine-rich tail (yellow), two transmembrane helices (green) and the periplasmic region (blue, salmon and ochre). **b** A schematic representation of the structure of Wzc$^{K540M}$ coloured as **a** The first residue of each structural block is shown. **c** Wzc$^{K540M}$ forms an octamer. The transmembrane helices are not close-packed and create portals to a large central cavity, clearly visible in the space fill representation. The structure has both a periplasmic and a cytoplasmic ring. **d** The Wzc$^{K540M}$ octamer viewed from the periplasm, reveals the periplasmic region also forms a ring-like arrangement. The central cavity is open to both the cytoplasm and lipid bilayer. Residues 65 to 84 could not be located experimentally and may occlude the entrance to the cavity from the periplasm. **e** The high quality of the EM map is illustrated by the holes for the aromatic residues. **f** The C-terminal tail from one monomer (coloured yellow and red) has residues labelled in normal text. Y717 is at the active site of the kinase domain from the other monomer (coloured cyan). Key kinase residues in the other monomer are labelled in bold italics. ADP and Mg$^{2+}$ are shown and labelled. **g** With Y717 at the active site, Y715 sits in a pocket where it makes a hydrogen bond with E675 from the neighbouring subunit. A phosphorylated tyrosine at position $i$ would be disfavoured by size and charge. Disruption of binding at position $i$ could perturb binding at position $i + 2$, the active site. Thus phosphorylation of Y715 seems most likely to follow (not precede) phosphorylation of Y717. Residues from the kinase domain are labelled in italics.

C-terminal kinase domain (Arg 452-Lys 701), and the tyrosine rich C-terminal tail (Ala 702-Lys 721) (Fig. 2b). The fold of the kinase domain was previously described in the crystal structure of the isolated cytoplasmic region from the closely related Wzc homolog from the colanic acid EPS biosynthesis system in *E. coli* (PDB 3LA6)[16], the kinase domain from *E. coli* Etk[18], and CapB from *S. aureus*[17]. Briefly, the kinase domain consists of central β-sheets sandwiched between two bundles of α-helices. The C-terminal tyrosine-rich tail (residues 707–721) reaches across into the other neighbouring subunit, similar to the previous crystal structure of the isolated kinase domain[16]. The kinase active site has been described in detail for Wzc[16], Etk[18] and CapB[17]. The ADP complex confirms the active site location (Supplementary Fig. 4e) with key catalytic residues conserved (Supplementary Fig. 5a). Residues 418 to 474 of Wzc are not found in *S. aureus* CapB, but instead are part of CapA, the corresponding membrane-embedded partner needed to activate the kinase[17]. The C-terminal tail begins with α-helix (α23) (Ala 702 to Arg 707) which packs against the N-terminal α1 helix (Fig. 2a, b). The conserved tyrosine-rich region (Tyr 708, 713, 715 and 717) (Supplementary Fig. 5a) reaches into the active site of the neighbouring subunit with Y717 positioned at the active site (Fig. 2f). Y715 sits in a pocket, where it makes a hydrogen bond with E675 from the neighbouring subunit site (Fig. 2g). In contrast, the crystal structure of the isolated kinase domain[16] has Y715 at the active site and Y713 is found in the pocket shown in Fig. 2g. In this crystal structure, the region corresponding to Ala 702-Arg 707 adopts a loop, rather than helix α23 seen in Wzc, possibly because the missing α1 helix is needed to stabilize α23. In CapB, the loop between β20 of the kinase domain and the shorter tyrosine-rich tail adopts a very different arrangement[17].

The transmembrane region of each Wzc monomer comprises two helices (α2 K31 to L50 and α12 I428 to F447), which form an X-like arrangement with multiple side-chain interactions between them (Fig. 2a). Together, the α12 helices create an inner ring in the octamer, while the eight α2 helices form an outer ring. The resulting arrangement of the eight transmembrane helical pairs is unusual; they are not closely packed, and this creates oval-shaped portals (20 Å wide by 45 Å high) located between the pairs of the helices (Fig. 2c).

The periplasmic region can be sub-divided into three distinct structural motifs (Fig. 2a, b). Motif 1 is formed by three non-contiguous stretches of the protein sequence (54–109, 209–267, 403–427) (Fig. 2b) and comprises four β sheets and four α helices (Fig. 3a, Supplementary Fig. 6a). The structure suggests a periplasmic pore in Wzc but caution is required concerning this interpretation because it could conceivably be occluded by the conserved stretch of residues 65–84, which are not located in the structure (Figs. 2d and 3b). The periplasmic ring principally creates contacts between motif 1 from each monomer (Fig. 3b). R410 of motif 1 is found at the interface, where it is within 4 Å of

the side chains of Q258 and R262 from the neighbouring monomer (Fig. 3c). R410 corresponds to R279 of Wzz (PDB 6RBG)[5] (Fig. 3d). Motif 2 (110–206) is a jellyroll of 8 antiparallel strands with one α helix (Fig. 3a, Supplementary Fig. 6a). In the octamer, motif 2 is positioned radially and makes no contact with the neighbouring monomers (Fig. 3b). Within each monomer, α5 packs against the transmembrane helix α2 (Fig. 3a).

In contrast to motifs 1 and 2, motif 3 (268–402) is not well ordered in the C8-averaged map (Supplementary Fig. 3f, g) and working in C1 identified three discrete structural arrangements (Fig. 3b, Supplementary Fig. 7a–e). This is an unusual break in symmetry. We designate the three possible arrangements as class i, ii and iii. In the most common particle, class i is found in three monomers (chains A, D and F) with all motif 3 residues visualised; this permits an almost complete model of the Wzc monomer (Supplementary Table 1, Fig. 2a, Supplementary Fig. 7a–c). Class i is a four coiled-coil helical bundle (α8–α11), composed of two helical pairs (α8–α11 and α9–α10). The helical axes are arranged perpendicular to the central eightfold axis of Wzc and are folded down onto the surface of motif 1 (Figs. 2a and 3b). Only a portion of α8 and α11 could be traced in class ii (chains B, C, G and H), where the helical axes are arranged approximately parallel to the eightfold axis (Fig. 3a, b and Supplementary Fig. 7a, b, d–f). This arrangement is found as a pair in neighbouring subunits, with the helices from one subunit stacking against the other (Fig. 3b). We were unable to identify any density for motif 3 in chain E (class iii) (Fig. 3b, Supplementary Fig. 7b). The striking differences in motif 3 point to conformational flexibility.

Approximately two-thirds of the surface area buried by the formation of octamer occurs in the cytosolic region (Fig. 3e), with the remainder arising from contacts between motif 1 of the periplasmic domain (Fig. 3b). The interaction between the C-terminal tyrosine-rich tail and the kinase active site of each neighbouring subunit (Fig. 3e) buries around 440 Å$^2$ surface area (of total 1200 Å$^2$ buried).

The transmembrane helices and most of motif 1 in Wzc can be superimposed with Wzz (PDB 6RBG)[5] with a root mean square deviation (rmsd) 2.1 Å over 130 Cα atoms (Fig. 3f). Wzz lacks the kinase domain and periplasmic motif 2. In Wzz, motif 3 is replaced by an extended helical bundle with a very different arrangement (Fig. 3f). The helical barrel evident in the Wzz octamer[5–7] is not found in Wzc (Supplementary Fig. 6b). Wzz and Wzc share the open arrangement of the transmembrane helices and the periplasmic ring formed by motif 1 (Supplementary Fig. 6b, c). However, the interfaces between motif 1 in Wzz bury approximately twice as much surface area as the corresponding motif in Wzc. In Wzc, the presence of motif 2 changes the local structure of motif 1 at the motif 1 contact interface; in Wzz this region makes extensive intersubunit contacts[5–7]. The unfavourable R410-R262 interaction of Wzc is

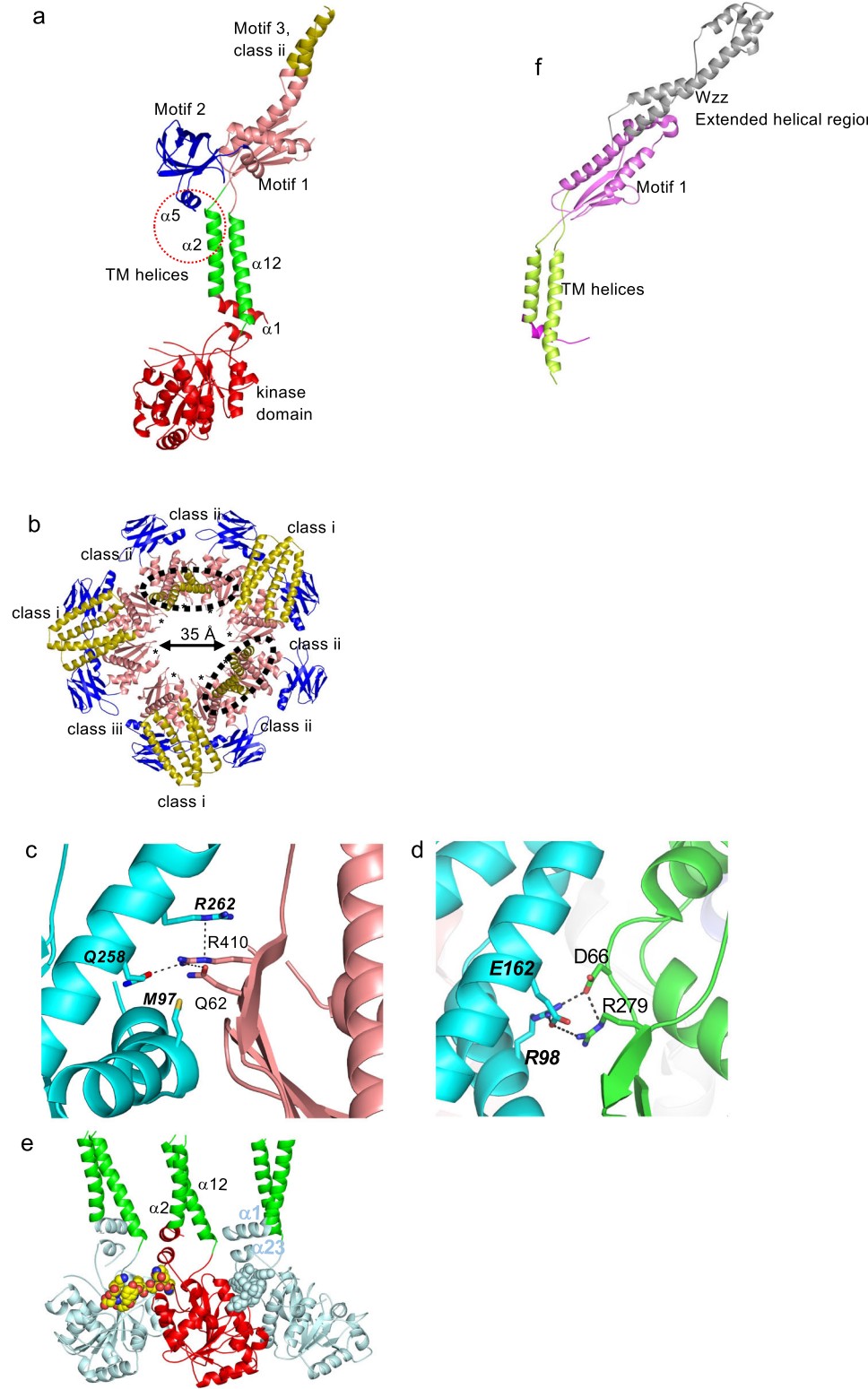

also present in Wzz (R279- R98) but there are compensating salt bridges with D66 and E162 in Wzz (Fig. 3d).

**The role of the tyrosine-rich tail.** In mass spectrometry, the native protein was shown to be phosphorylated and predominantly monomeric, echoing reports of the behaviour of isolated kinase domains[18] (Fig. 1b). Systematic Tyr to Phe

mutations established that the C-terminal tyrosine residues were essential for capsule production, but no single tyrosine was essential, nor was any single tyrosine sufficient[14,24]. Reasoning the negatively charged Glu would mimic negatively charged pTyr (phosphotyrosine) and create a protein resembling a locked phosphorylated state, several Tyr-to-Glu mutants were constructed in Wzc$^{K540M}$. With the exception of Y706E, each single Tyr to Glu replacement, as well as double (717 and 718) and triple

**Fig. 3 The periplasmic and cytoplasmic rings of Wzc. a** The class ii arrangement of motif 3 is shown. This is very different to the class i arrangement shown in Fig. 2a, suggesting motif 3 is dynamic. Motif 2 makes contact with $\alpha_2$ thereby forging a structural connection between the periplasm and cytoplasm. The colour scheme is as Fig. 2a. **b** The periplasmic ring of Wzc, using the colour scheme in Fig. 2a. The ring is predominantly held together by contacts between motif 1 of neighbouring monomers. The class ii arrangement is found as a pair where the helices from one monomer stack with the other monomer (circled). In the class iii arrangement, the helices are almost entirely disordered. **c** R410 is at the interface between the periplasmic domains of Wzc, where it makes interactions with Q62 and from the neighbouring subunit R262 and Q258 (bold italics). Interactions closer than 4 Å are shown as dashed lines. The interface in Wzz is shown in (**d**). **d** The interface in Wzz (PDB 6RBG)[5], corresponding to that shown for Wzc in (**c**). In Wzz, R279 (equivalent to R410 in Wzc) is at the interface between the periplasmic domains, where it makes interactions with D66 and from the neighbouring subunit R98 and E162 (denoted bold italics). Interactions close to 4 Å are shown as dashed lines. **e** The cytoplasmic ring of Wzc$^{K540M}$ is held together by interactions between the neighbouring kinase domain and the tyrosine-rich C-terminus which is a target for phosphorylation (shown as a space fill sphere) that reach the active site of the neighbouring monomer. **f** Full-length Wzz (PDB 6RBG)[5], shares the same arrangement of TM helices (pale green) and part of motif 1 (pale pink). The extended helical region (grey) in Wzz does not resemble any class of motif 3 in Wzc. Thus, the kinase domain, motif 2 and motif 3 of Wzc are all unique features. The octameric assembly of Wzz is shown in Supplementary Fig. 6b, c.

(715, 717 and 718) mutants formed octamers, as judged by EM (Supplementary Fig. 8a). We term this set of Y to E mutants "octamer capable". A crude analysis of the percentage of octamers on the EM grid shows that as the number of Y to E substitutions increased, the portion of particles that were octameric reduced (Supplementary Fig. 8a). The capacity of these mutants to sustain autophosphorylation and polysaccharide production was assessed by constructing the same mutations in native (phosphorylation-proficient) Wzc protein. As expected, western blotting and mass spectrometry showed the amount of phosphorylation decreased with increasing numbers of replaced tyrosine residues (Fig. 4b, Supplementary Fig. 2c). Mass spectrometry showed these mutants are found as (with decreasing extent) phosphorylated monomers (Supplementary Fig. 2c), similar to the native protein. The "octamer capable" Y to E mutants (Supplementary Fig. 8a) all supported capsule production in vivo when introduced into the native *wzc* background (Supplementary Fig. 8b, c). Structural analysis suggests the Y706E mutant would perturb the interaction of $\alpha_1$ and $\alpha_{23}$, consistent with the observed functional impairment in single and double mutants involving Y706.

The introduction of four or more Glu residues resulted in severe defects in capsule production (Fig. 4a, Supplementary Fig. 8f). Phosphorylation of these proteins was not detected in whole-cell lysates (Fig. 4a) but this was attributed to reduced levels and limits of detection in lysates; phosphorylation was confirmed with the purified proteins (Fig. 4b, Supplementary Fig. 2c). A mutant with seven Glu residues did not support capsule production or autophosphorylation, as predicted by previous work[24] (Supplementary Fig. 8f).

Analysis of cryo-EM images of Wzc$^{K540M}$ with four Glu residues (designated Wzc$^{K540M}$4YE) revealed ~50% octamers, as well as other state(s) (Fig. 4c, Supplementary Fig. 9b–d). A 2.8 Å-resolution structure of the Wzc$^{K540M}$4YE octamer was determined using the same approach as Wzc$^{K540M}$ (Supplementary Fig. 9a–j). Overall, this structure has similar octameric (rmsd 3.1 Å over 4520 Cα atoms) and monomeric (rmsd of 2.5 Å over 651 Cα atoms) arrangements seen in Wzc$^{K540M}$ (Supplementary Fig. 9e). The core kinase domain is unchanged between the structures (rmsd 0.4 Å for Cα atoms 452–697) but the tyrosine-rich tail has undergone a large change that now places Y708 at the active site (compared Y717 in Wzc$^{K540M}$) (Fig. 4d). Excluding the C-terminal tail and re-calculating the superpositions for monomer and octamer yields rmsd values of 1.8 Å and 2.3 Å respectively. The change in the tail results in a reduction of the buried surface area that holds the octamer together. In Wzc$^{K540M}$4YE, Y706 points into the same pocket as Y715 in Wzc$^{K540M}$ (Fig. 2g) but this residue no longer makes the hydrogen bond with E675. In Wzc$^{K540M}$4YE, helix $\alpha_{23}$ is unwound, and the resulting loop no longer makes any contacts with helix $\alpha_1$, whose position is altered as a result (Fig. 4d). The

structure supports a model where the Tyr residues located C-terminal to Y708 have been phosphorylated, consistent with the observed multiple phosphorylation of the C-terminal peptide[13–15].

The secondary structures of motif 1 and motif 2 in the periplasmic region are preserved in Wzc$^{K540M}$4YE but the structure comparison reveals some changes (rmsd 0.8 Å for 207 Cα atoms). Motif 3 still possesses the same three structural arrangements seen in Wzc$^{K540M}$ but only two (not three) monomers have the fully ordered class i arrangement (rmsd of 0.4 Å for 133 Cα atoms). The larger rmsd value for the monomer and octamer, compare to the individual regions, reflects shifts in the relative positions of the regions. This can be visualised by superimposing the core kinase domains of monomer for Wzc$^{K540M}$ and Wzc$^{K540M}$4YE, which shows a rigid body rotation of 11° at the other end of the molecule (Fig. 4e). Superposition using all the Cα atoms in octamer reveals shifts in the periplasmic helices, particularly those in class ii (Fig. 4f).

pY718 was observed in some peptides of the native protein (Supplementary Fig. 2a) but it is not a conserved residue (Supplementary Fig. 5a). An additional set of multiple Tyr to Glu mutants was constructed in order to test whether Y718 itself played a role and revealed that Y718 has no functional significance (Supplementary Fig. 8a, d, e).

**The role of motif 3.** The structures of the helical bundles appear to be dynamic within the octamer (resulting in three possible arrangements). Comparing Wzc$^{K540M}$ vs Wzc$^{K540M}$4YE revealed the distribution of arrangements changed, suggesting the helical bundles are particularly sensitive to changes elsewhere in the structure. However, we are unable to exclude the possibility that this difference is due to a relatively smaller dataset of Wzc$^{K540M}$4YE compared with Wzc$^{K540M}$. To further probe the functional significance of the helical bundles, a construct lacking motif 3 (ΔMotif3) was engineered in either Wzc or Wzc$^{K540M}$ backgrounds. As expected, purified Wzc$^{K540M}$ΔMotif3 formed octamers, same as Wzc$^{K540M}$ (Fig. 5a). However, the WzcΔMotif3 protein was unable to support capsule production establishing these helices are key to function, despite the preservation of its autokinase activity (Fig. 5b).

## Discussion

Wzc participates in the production of critical CPS virulence factors in prominent multidrug-resistant ESKAPE pathogens *Klebsiella pneumoniae*[25] and *Acinetobacter baumannii*[26]. This makes Wzc a candidate for antibacterial development to counter these and other pathogens. In the *E. coli* K30 prototype, Wzc (PCP-2a) proteins regulate polymerization, as deletion of *wzc* results in no accumulation or synthesis of any high-molecular-

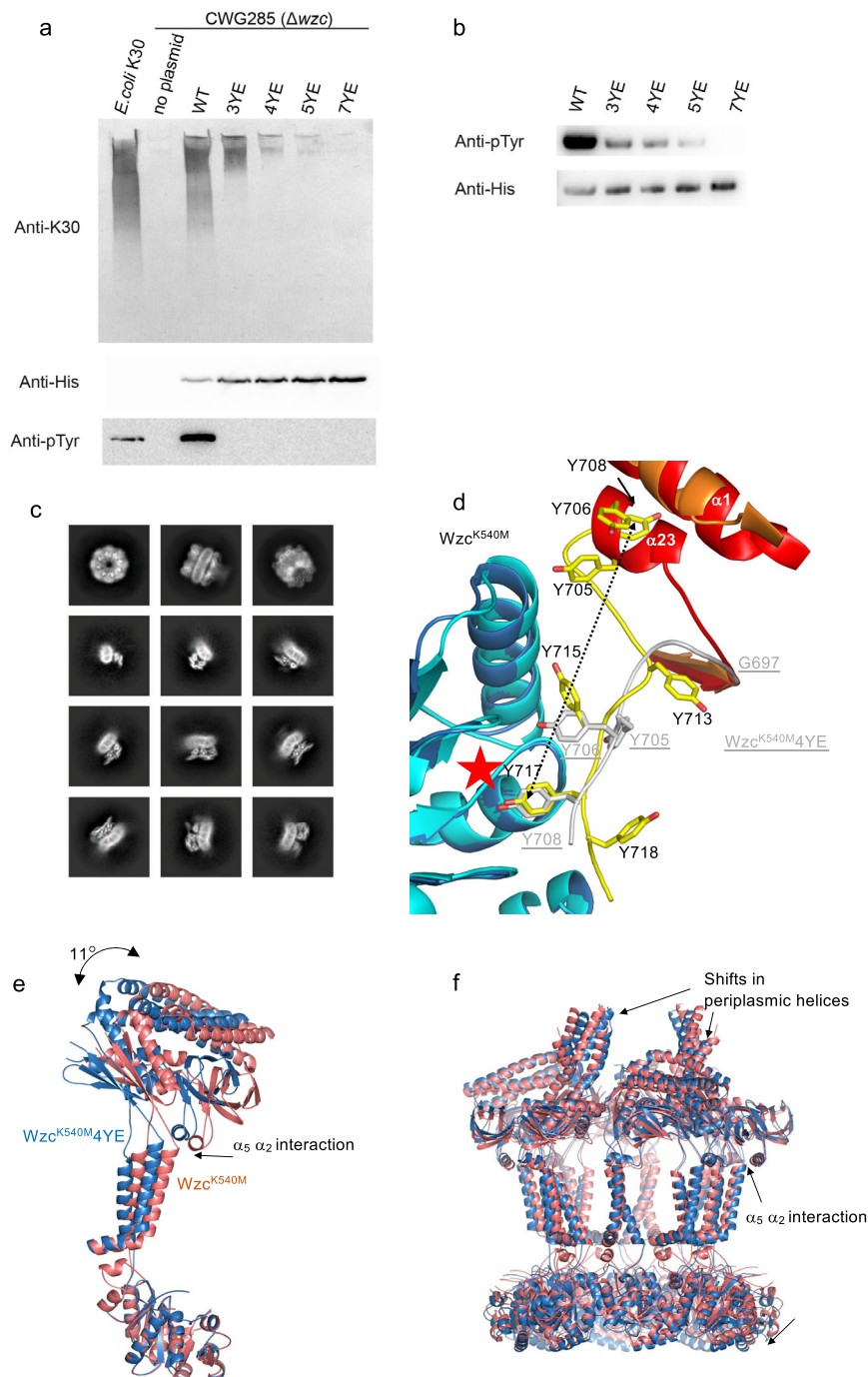

weight CPS[1,13]. An explicit link between Wzc and polymer chain length was also reported for *E. coli* colanic acid[14]. A gain-of-function mutation in *A. venetianus* RAG1 *wzc*, (equivalent to R410L in Wzc) leads to a hyper mucoid phenotype and increased apparent molecular weight (and potential value) of the commercial biopolymer known as emulsan[27].

Since Wzc and its O-antigen biosynthesis counterpart, Wzz[5], both regulate the polymerase Wzy, this aspect of their function(s) would be expected to reflect some structural conservation. Both proteins share most of motif 1 which forms an octameric ring and both have a large chamber in the membrane formed by the transmembrane helices (Supplementary Fig. 6b, c). The chamber offers a plausible location for Wzy and the portals would allow

free exchange of lipid-linked oligosaccharides substrates. This location would imply contact between Wzy and $\alpha_{12}$ of Wzc and some experimental support for this proposal comes from mutants of the equivalent helix of Wzz that are known to alter Wzy behaviour[28].

Wzc possesses a cytoplasmic autokinase domain and two periplasmic sub-domains (motifs 2 and 3) that are absent in Wzz. Non-phosphorylated Wzc is an octamer held together by the interaction between tyrosine-rich C-terminal peptide of one monomer and the active site of the kinase domain of neighbouring monomer (Figs. 2c, 3e). The structure of Wzc$^{K540M}$4YE, chosen to mimic a phosphorylated state, showed a significant conformational change in the tyrosine tail, which would be

**Fig. 4 The effect of phosphorylation. a** Western immunoblot of whole cell lysates probed with anti-K30 antiserum to detect cell surface polysaccharides (top panel), the anti-His antibody for expression (middle panel) and anti-pTyr antibody for pTyr (bottom panel). *E. coli* K30 (strain E69) is used as a positive control. *E. coli* CWG285 (*wzc* mutant) is used as a negative control. WT, 3YE, 4YE, 5YE and 7YE represent plasmids encoding wild-type Wzc, Y715E/Y717E/Y718E mutant, Y713E/Y715E/Y717E/Y718E mutant, Y708E/Y713E/ Y715E/Y717E/Y718E mutant and Y705E/Y706E/Y708E/Y713E/ Y715E/Y717E/Y718E mutant respectively. Bacteria were grown for 16 h without L-arabinose induction to obtain Wzc expression levels similar to chromosomal copy. Uncropped gels with molecular weight markers are shown in Supplementary Fig. 8f. Experiments were performed in biological triplicate and were consistent, a representative experiment is shown. **b** Western immunoblot of purified Wzc proteins probed with anti-pTyr antibody and anti-His antibody is labelled as above. The phosphorylation of 3YE,4YE,5YE not visible in (**a**) is attributed to sensitivity. Uncropped gels with molecular weight markers in shown in Supplementary Fig. 10a. Experiments were performed in technical triplicate. **c** Representative 2D class averages of Wzc$^{K540M}$4YE (mimicking phosphorylation of tyrosines) showing octamer and non-octameric particles (box size 312 Å). **d** A superposition of the kinase active site of Wzc$^{K540M}$ (coloured as Fig. 2f) and Wzc$^{K540M}$4YE (the N-terminal helix α1 is coloured in orange, the C-terminal tail in grey and the residues in sticks shown with carbons white, residues are labelled in underlined grey). The C-terminus of Wzc$^{K540M}$4YE has translocated through the active site (red star), the shift in Y708 is shown as an arrow. As a consequence of the movement, α23 has been entirely unwound which in turn perturbs α1. **e** Superposition (using the kinase domain) of the monomers of Wzc$^{K540M}$ (coloured salmon) and Wzc$^{K540M}$4YE (dark blue) shows a structural shift in the protein with a relative rotation in the periplasmic domains, these changes are transmitted through the transmembrane helices. **f** Superposition of the octamer of Wzc$^{K540M}$ (coloured salmon) and Wzc$^{K540M}$4YE (dark blue) reveals shifts in the periplasmic helices.

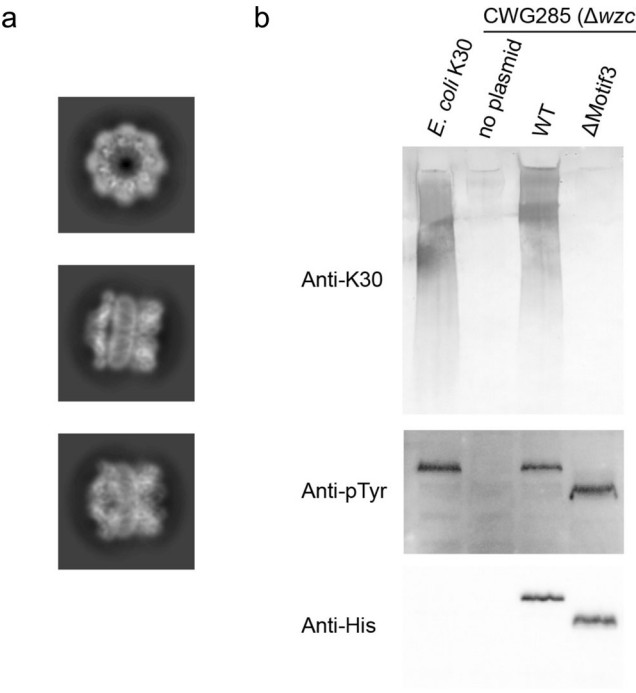

**Fig. 5 Function of Wzc periplasm motif 3. a** Representative 2D class averages of Wzc$^{K540M}$ ΔMotif3 show the protein forms an octamer. Cryo-EM data were collected on Glacios (Thermo Fisher). Box size: 248 Å. **b** Western immunoblot of whole cell lysates probed with anti-K30 antiserum to detect cell surface polysaccharides (top panel), the anti-pTyr antibody for pTyr (middle panel) and anti-His$_5$ antibody for expression (bottom panel). Bacteria were grown for 16 h without L-arabinose induction to achieve amounts of Wzc similar to those from chromosomal copies. Uncropped gels with molecular weight markers are shown in Supplementary Fig. 10b. Experiments were performed in biological triplicate and were consistent, a representative experiment is shown.

predicted to weaken the octamer (Fig. 4c, d, Supplementary Fig. 9b–d). There is some evidence for different oligomeric states (other than octamer) in Wzc$^{K540M}$4YE (Supplementary Fig. 9d) but none were amenable to structural determination. The introduction of one further replacement (5YE) resulted in the complete loss of the octamer form (Supplementary Fig. 8a).

Systematic mutagenesis indicated that progressive phosphorylation destabilises the octamer (Supplementary Fig. 8a) until the point where four conserved tyrosines (Y708, Y713, Y715, Y717) are phosphorylated. This species represents the tipping point where the stable octamer is unable to form and the protein found as monomers. Structural data would suggest phosphorylation typically starts at Y717 (Fig. 2f). In Wzc$^{K540M}$, Y715 sits in a pocket (Fig. 2g) that would disfavour binding of a larger negatively charged pTyr. Thus Y717 seems highly unlikely to be phosphorylated after Y715, since if pY715 cannot enter this pocket then the correct positioning of Y717 at the active site would also be disrupted (Fig. 2f). A similar logic applies when considering Y713 and Y715, phosphorylation of Y713 follows phosphorylation of Y715. The most abundant 5 P and 4 P peptides possess pY708, pY713, pY715, pY717, in a pattern predicted by structural analysis to arise from sequential C-terminal phosphorylation. This is consistent with mass spectrometry (Supplementary Fig. 2a). As the peptide passes through the active site, the surface area buried by the interaction of the tyrosine-rich peptide and the kinase domain progressively decreases, destabilising the octamer. The Wzc$^{K540M}$4YE structure shows that the tyrosine-rich tail would be accessible to the cognate phosphatase Wzb (Fig. 4d)[9,29,30]. Wzc would be expected to constantly cycle between phosphorylated monomers and dephosphorylated octamers in the cell, consistent with the requirement for both kinase and phosphatase activity[13–15].

Several proteins in the synthetic machinery of Gram-positive bacteria have been reported to be activated by PCP-2b-mediated phosphorylation (reviewed in[1,31]). Tyrosine phosphopeptides have been identified in enzymes involved in the Gram-negative *Klebsiella* capsule biosynthesis[32], and there are differing reports on Wzc-mediated phosphorylation of the biosynthetic uridine diphosphoglucose dehydrogenase in *E. coli*[33,34]. This suggests that monomeric Wzc, with its accessible active site, may exert additional regulatory effects upon bacterial metabolism.

Available data indicates that Wzc engages with Wza during export and synthesis. This evidence can be summarised as (i) genetic experiments support cognate recognition of Wza-Wzc partners[21]; (ii) Wza-Wzc heterocomplexes have been observed by microscopy[22]; (iii) deletion of *wza* prevents polymer production in a phenotype identical to *wzc*-deletions, suggesting an indirect (feedback) on the Wzy polymerase[35]. In purified Wza octamers, the OM channel is closed to the periplasm by a periplasmic tyrosine ring[19,36] and this must be opened for translocation as the polysaccharide passes through its central cavity[20]. Structural analysis of Wzc identifies the large periplasmic helical bundle (motif 3) (Fig. 2a) as the candidate for interaction with the periplasmic portion of octameric translocase Wza. Motif 3 is absent in Gram-positive bacteria (Supplementary Fig. 5b) which, of course, lack an OM. In addition, Wzz (which lacks an OM protein partner) has an entirely different helical structure from

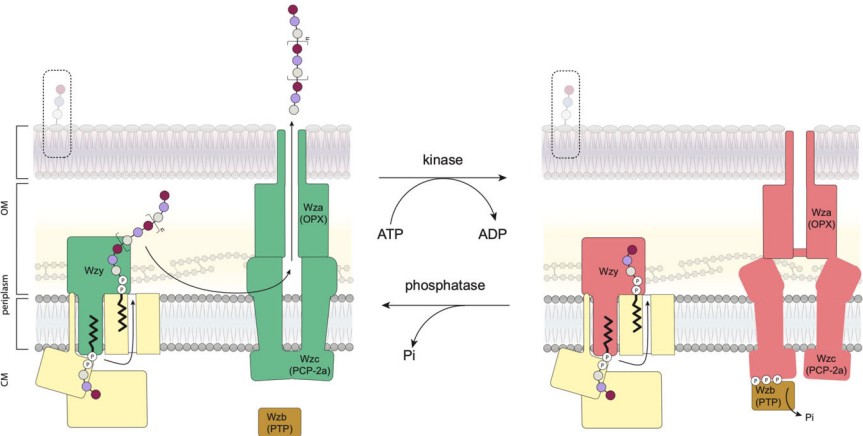

**Fig. 6 Model for Wzc function.** Wzc, the master regulator of capsule assembly and export regulates the activity of Wzy (polymerase) and opens Wza (translocon). Multi phosphorylated Wzc is found as a monomer whilst dephosphorylated Wzc forms an octamer. Capsule product and export requires continuous Wzc cycling between phosphorylation and dephosphorylation states. Polymer chain completion may be accomplished in a single or stepwise by multiple cycles.

Wzc motif 3 (Fig. 3f). Deletion of the periplasmic helices in motif 3 resulted in a Wzc variant protein that retained the ability to form octamers and possessed normal autokinase activity (Fig. 5a, b). However, the mutant abolished the production of polysaccharides uncoupling these activities from phosphorylation (Fig. 5b). This phenotypic outcome resembles deletion of Wza in vivo[35]. The data is consistent with these helices being key to the control of Wza and the interaction of Wzc with Wza being required for activation of Wzy.

We propose that the machine that drives synthesis and export of EPS is a ternary complex minimally comprising octameric Wzc, octameric open-form Wza and Wzy. Autophosphorylation of Wzc results in octamer dissociation, leading to the dissolution of the machine which halts polymerisation and closes Wza (Fig. 6). The capacity to reassemble the machine for renewed synthesis and export, therefore, depends on the dephosphorylation of Wzc and rationalises the essential requirement for Wzb.

The molecular insights into this system provide the essential foundation to investigate these predictions and offer opportunities for efforts to target polysaccharide biosynthesis for both therapeutic and industrial benefit.

## Methods

**Cloning, expression and purification of Wzc.** The *wzc* gene from *E. coli* E69 (O9a:K30) gene was cloned into vector pBAD24[37] to generate pBRC901, in order to express Wzc with a C-terminal hexa-histidine tag. *E.coli* TOP 10 cells transformed with pBRC901 were grown at 37 °C until $OD_{600}$ reached approximately 0.8, then Wzc expression was induced with 0.002% arabinose at 20 °C overnight. Cells were harvested by centrifugation and stored at −80 °C. Cell pellets were resuspended with lysis buffer (20 mM Na phosphate, pH7.0, 500 mM NaCl) and lysed by passage through a Constant Systems cell disruptor. Unbroken cells were removed by centrifugation at 20,000×g for 1 h at 4 °C. Membranes were collected by ultracentrifugation at 186,000×g for 1 h at 4 °C and solubilized with 20 mM Na phosphate, pH7.0, 500 mM NaCl, 1% DDM at 4 °C. Residual membrane debris was removed by ultracentrifugation and the supernatant containing Wzc was purified under gravity using ABT nickel resin (Cat. No. 6BCL-NTANi-100). The protein was eluted with 20 mM Na phosphate, pH 7.0, 500 mM NaCl, 0.003 % Lauryl Maltose Neopentyl Glycol (LMNG) containing 300 mM imidazole after successive column washes with 20 mM imidazole and 50 mM imidazole. Purified protein was buffer exchanged to buffer (20 mM Na phosphate, pH 7.0, 500 mM NaCl, 0.003 % Lauryl Maltose Neopentyl Glycol (LMNG)) using a CentriPure P100 column and concentrated with a 100 kDa concentrator. Concentrated protein was applied to a Superose 6 10/300 increase column equilibrated with 20 mM HEPES, 150 mM NaCl, 0.001 % LMNG, 2 mM tris(2-carboxyethyl)phosphine (TECP), pH 7.3. Wzc mutant derivatives were made through site-directed mutagenesis or by Gibson assembly following the manufacture's protocols (New England Biolabs). Expression and purification of Wzc mutants were performed as described for the native protein.

For the dephosphorylation, Wzc was incubated with Wzb[9] at room temperature in a molar ratio of 1:5. Samples were taken after 0, 1 h, 2 h and 3 h intervals and the presence of phosphorylation on tyrosine were monitored as described below.

**Western immunoblot detection of phosphorylated Wzc proteins.** In total, 8 µl of 0.2 mg/ml purified proteins were analyzed by western immunoblotting using mouse monoclonal anti-pTyr antibody (Sigma, Cat. No. P4110, dilution 1:5000) as primary and an HRP conjugated anti-mouse IgG antibody (Promega, Cat. No. W402B, dilution 1:5000) as secondary antibody. An anti-polyHistidine-peroxidase antibody (Sigma, Cat. No. A7058, dilution 1:5000) was used to detect hexa-histidine tagged Wzc protein, for showing the loading amount.

Wzc variants were also examined in whole-cell lysates prepared from *E. coli* CWG285 transformants grown in LB medium (containing 100 µg/ml ampicillin where required) at 37 °C. CWG285 is derived from *E. coli* E69 and contains one insertion in *wzc*, and another insertion that exerts a polar inactivating effect on the *wzc* homolog, *etk*[13]. Lysates were prepared by harvesting 1 $OD_{600nm}$ equivalent of bacteria by centrifugation at 12,000 × g for 2 min. The pellets were lysed in 100 µL of Laemmli buffer (125 mM Tris-Cl, pH 6.8, 20 % (w/v) glycerol, 4% (w/v) SDS, 0.004 % (w/v) bromophenol blue)[38] and heated at 100 °C for 10 min. The whole-cell lysates were analysed by SDS-PAGE and transferred to Protran 0.45 ìm nitrocellulose membrane (Amersham). Hexa-histidine-tagged Wzc was identified using mouse anti-pentaHis antibody (Qiagen, 1:2000) and phosphorylated Wzc was detected with mouse PY20 antibody (Sigma, Cat. No. P4110, 1:2000). Goat anti-mouse horseradish peroxidase (HRP) conjugated antibody (Cedar Lane, 1:3000) was used as the secondary antibody. HRP was detected by chemiluminescence using the Crescendo Western HRP substrate (Millipore). The uncropped images of gels are available in Supplementary Fig. 10.

**Western immunoblot analysis of cell surface polysaccharides.** Wzc variants were tested for function by transforming the corresponding plasmids into *E. coli* CWG285. Bacteria were grown and whole-cell lysates prepared as described above, with the exception that samples of CPS analysis were treated with proteinase K (0.5 mg/mL) at 55 °C for 1 h[39]. The whole-cell lysates were separated and transferred to nitrocellulose as described above. K30 CPS was detected using rabbit anti-K30 antiserum[40] as the primary antibody at a 1:3000 dilution and goat anti-rabbit alkaline phosphatase (AP) conjugated antibody (Cedar Lane, 1:3000) as the secondary antibody. AP was detected with nitro blue tetrazolium and 5-bromo-4-chloro-3-indolyl phosphate (Roche).

**Native mass spectrometry.** Prior to MS analysis, the samples were buffer exchanged into 200 mM ammonium acetate pH 8.0 and 2× critical micelle concentration of detergent (C8E4 and LDAO) using Biospin-6 (BioRad) column and introduced directly into the mass spectrometer using gold-coated capillary needles (prepared in-house). Data were collected on a Q-Exactive UHMR mass spectrometer (ThermoFisher). The instrument parameters were as follows: capillary voltage 1.2 kV, quadrupole selection from 1000 to 20,000 m/z range, S-lens RF 100%, collisional activation in the HCD cell 200–300 V, trapping gas pressure setting 7.5, temperature 250 °C, resolution of the instrument 12,500. The noise level was set at 3 rather than the default value of 4.64. No in-source dissociation was applied. Data were analyzed using Xcalibur 4.2 (Thermo Scientific) software package. All experiments were repeated three times with similar outcomes.

**Phosphoproteomics and phosphopeptide analysis**. For phosphorylation identification, the tryptic peptides were loaded onto a reverse phase C18 trap column (Acclaim PepMap 100, 75 µm × 2 cm, nano viper, C18, 3 µm, 100 Å, ThermoFisher, Waltham, MA, U.S.A) using an Ultimate 3000 and washed with 50 µL of 0.1 % formic acid at 10 µl/min. The desalted peptides were then separated using a 15 cm pre-packed reverse-phase analytical column (Acclaim PepMap 100, 75 µm × 15 cm, C18, 3 µm, 100 Å, ThermoFisher, Waltham, MA, U.S.A) using a 45 min linear gradient from 5 to 40% eluant (80% acetonitrile, 20% water, 0.1% formic acid) at a flow rate of 300 nL/min. The separated peptides were electrosprayed into an Orbitrap Eclipse Tribrid mass spectrometry system in the positive ion mode using data-dependent acquisition with a 3 s cycle time. Precursors and products were detected in the Orbitrap analyzer at a resolving power of 60,000 and 30,000 (@ m/z 200), respectively. Precursor signals with an intensity $>1.0 \times 10^{-4}$ and charge state between 2 and 7 were isolated with the quadrupole using a 0.7 m/z isolation window (0.5 m/z offset) and subjected to MS/MS fragmentation using higher energy collision-induced dissociation (30% relative fragmentation energy). MS/MS scans were collected at an AGC setting of $1.0 \times 10^4$ or a maximum fill time of 100 ms and precursors within 10 ppm were dynamically excluded for 30 s. Raw data files were processed using MaxQuant (version 1.6.3.4) and searched against the *Escherichia coli* (O9a:K30) biosynthesis gene cluster[30] as well as a list of common contaminants using the Andromeda search engine[41,42] with the following search parameters: trypsin digestion; fixed modification was set to carbamidomethyl (C); variable modifications set as oxidation (M), acetylated protein N terminus, and phosphorylation (Y); up to two missed cleavages allowed. Mass spectra were recalibrated within MaxQuant with a precursor error tolerance of 20 parts per million (ppm) and then re-searched with a mass tolerance of 5 ppm. Note that our initial analyses with pSYTH search yielded the assignment of phosphosites on His and Ser along with the conserved Tyr residues within the peptide 708–718 given their close proximity, however, based on manual inspection and phosphoproteomics analyses on several mutants (data is on PRDIE with accession number PXD025820) suggested that the assignment on His and Ser was not correct. Thus, we used pY only for searching phosphosites in our subsequent analyses.

**EM analysis**. For negative stain, 3.5 µl of purified protein (around 0.05 mg/ml) was applied to glow-discharged 400-mesh carbon-coated copper grids, which were then washed three times with ddH$_2$O and stained with 0.75% uranyl formate. Images were acquired using an FEI Tecnai T12 microscope at a magnification of ×42,000, with a calibrated pixel size of 2.63 Å.

For cryo-EM of Wzc$^{K540M}$ and Wzc$^{K540M}$4YE, 3.5 µl purified sample at ~2 mg/ml was applied to glow-discharged Quantifoil gold R1.2/1.3 300-mesh grids and the grids were blotted for about 3 s in conditions of 100% humidity and 4 °C before vitrification in liquid ethane using Vitrobot (FEI). All datasets were collected on a Titan Krios equipped with a K3 direct electron detector at eBIC, Diamond Light Source, UK. The apo- and ADP-complex datasets were collected by SerialEM at a magnification of ×105,000 with the physical pixel size of 0.829 Å/pixel. The total dose was 55 e/Å$^2$ and 53.6 e/Å$^2$ for apo- and ADP-complex datasets, respectively, and the defocus range was −0.5 µm to −2.5 µm. The 4YE dataset was collected by EPU software at a magnification of ×105,000, with the physical pixel size 0.831 Å/pixel, using a defocus range of −0.5 µm to −3.0 µm. The total dose was 57.5 e/Å$^2$.

All EM movies were motion-corrected by MotionCor2[43] through Relion3.0. Contrast transfer function values were estimated by Gctf[44] or CTFFIND[45]. Particles were picked using Laplacian-of-Gaussian (LoG) based auto-picking in Relion3.0[46]. After extraction and normalization in Relion3.0, particles were imported into cryoSPARC[47] for further processing. Multiple rounds of 2D classification were carried out. The initial model was generated from a selection of 2D classes in cryoSPARC[47]. Other rounds of 3D classifications were carried out without applying symmetry. C8 symmetry was observed for the resulting maps, except motif 3. For the ADP-complex and 4YE mutant maps, the best 3D classes were reconstructed using non-uniform refinement[48] in cryoSPARC applying C1 or C8 symmetries. For the apo-structure, after cleaning through multiple rounds of 2D and 3D classification, 673,112 particles were used for further 3D classification with C1 symmetry. One representative class with the most complete motif 3 was chosen for non-uniform refinement to generate a 2.85 Å map at C1 symmetry. A mask covering only the periplasmic region was generated and local refinement was carried out in cryoSPARC[47]. The resulting 2.77 Å map showed clear density for the side chains of the periplasmic motif 3. For the C8 reconstruction, further 3D classifications were carried out and 546,088 particles were subjected to one final round of non-uniform refinement with C8 symmetry to yield a 2.3 Å map (Supplementary Fig. 3a).

For Wzb treated Wzc, 3.5 µl of purified sample at ~1.8 mg/ml was used to make the grids in the same way described above. In total, 1787 micrographs were collected on Glacios (Thermofisher) at a magnification of ×92,000 with the physical pixel size 1.55 Å/pixel. Defocus range was −1.0 µm to −2.5 µm. Total dose was 60 e/Å$^2$. Using convolutional neural networks based algorithm Topaz[23] we were able to pick 356,511 particles. Multiple rounds of 2D classifications were carried out, and the class averages showing clear Wzc octameric end views were selected and analysed.

Grids of Wzc$^{K540M}$ΔMotif3 at ~1.9 mg/ml were made in the same way described above. Dataset was collected on Glacios at a magnification of ×92,000

with the physical pixel size 1.55 Å/pixel. The total dose was 60.38 e/Å$^2$. Data was processed similar to Wzc$^{K540M}$ to get 2D class averages.

**Model building and refinement**. For the apo-structure, the cytoplasmic kinase domain was refined from a homology model (PDB code 3LA6). The other regions were built manually in Coot[49]. The 2.3 Å C8-symmetry map was used to build the structure, with the exception of motif 3. The locally refined 2.77 Å C1 symmetry map was used to build the three classes of motif 3. The entire structure was assembled in Coot and refined against the 2.85 Å C1 map. The structure without motif 3 was refined against the C8 map. For the ADP-bound complex structure, ligands were manually added in Coot and the structure was refined in PHENIX[50]. For the 4YE mutant structure, the mutation points were manually built in Coot, based on the apo-structure, and refinement was carried out in PHENIX. Figures of cryo-EM maps and structures were generated with PyMOL (The PyMOL Molecular Graphics System, Version 2.1 Schrödinger, LLC) and UCSF Chimera[51].

**Antibodies**. Monoclonal anti-polyHistidine-peroxidase antibody produced in mouse was purchased from Sigma (Cat. No. A7058). Monoclonal anti-phosphotyrosine antibody produced in mouse was purchased from Sigma (Cat. No. P4110). HRP conjugated anti-mouse IgG (H + L) antibody was purchased from Promega and used as the secondary antibody for detecting phosphotyrosine (Cat. No.W402B). AP-conjugated goat anti-rabbit IgG was purchased from Cedarlane (Cat. No. CLCC43008). Peroxidase-conjugated goat anti-mouse IgG was purchased from Cedarlane (Cat. No. 115-036-003). Penta-his antibody was purchased from Qiagen (Cat. No. 34660)

**Reporting summary**. Further information on research design is available in the Nature Research Reporting Summary linked to this article.

## Data availability

The data that supports this study are available from the corresponding authors upon reasonable request. EM maps and models are deposited in the EMDB and wwPDB under accession codes EMD-12338 and PDB 7NHR (C1 WzcK540M); EMD-12339 and PDB 7NHS (C8 WzcK540M); EMD-12340 (WzcK540M periplasmic localized map); EMD-12360 and PDB 7NII (C1 WzcK540M ADP complex); EMD-12359 and PDB 7NIH (C8 WzcK540M ADP complex); EMD-12353 and PDB 7NIB (C1 WzcK540M4YE); EMD-12349 and PDB 7NI2 (C8 WzcK540M4YE). The mass spectrometry proteomics data have been deposited to the ProteomeXchange Consortium via the PRIDE partner repository with the dataset identifier PXD025820. All constructs are available from the authors upon request until their deposition and release by ADDGENE. Source data are provided with this paper.

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

## Acknowledgements

J.H.N., Y.Y., and P.W. are supported by Wellcome Trust (100209/Z/12/Z). J.R.B. and C. V.R. are supported by MRC Programme grants MR/N020413/1 and MR/V028839/1 where J.R.B. is a Researcher Co-Investigator on the latter. We acknowledge the Wellcome Trust Membrane Protein Laboratory (20289/Z16/Z) for support. C.W. is the recipient of a Canada Research Chair and his research (and that of L.S. and B.R.C) is supported by a Canadian Institutes of Health Research Foundation grant (FDN-2016-148364 to CW). J. L. and P.Z. are supported by Wellcome Trust Investigator Award (206422/Z/17/Z). We acknowledge Diamond for access and support of the CryoEM facilities at the UK national electron bio-imaging centre (eBIC, proposal EM20223), funded by the Wellcome Trust, MRC and BBSRC. All grids pre-screened and selected using the cryo-EM facility (OPIC) in the Division of Structural Biology, University of Oxford, part of the UK Centre of Instruct-ERIC. The Franklin is a core funded research Institute of the EPSRC.

## Author contributions

Y.Y. purified proteins with contributions from P.N.W., Y.Y., J.L., P.Z., and J.H.N. carried out the E.M. structural studies and analysis. B.C., L.S., and C.W. carried out the in vivo assay and analysis of Wzc mutant function. J.B. and C.V.R. carried out the mass spectrometry. J.H.N. and C.W. led the study. All authors contributed to the analysis of data and writing of the manuscript.

## Competing interests

The authors declare no competing interests.
