## [Peer Review File · Nature Communications]

Reviewers' Comments:

Reviewer #1:

Remarks to the Author:

The authors present in this manuscript the near atomic resolution structure of an E.coli wzc polysaccharide co-polymerase involved in bacterial capsid assembly. There are a few structures of isolated kinase domains as well as a wzz structure describing the TM domain from this family of proteins; however this would be the first fully intact structure for this sub-class of PCP proteins. The authors accomplish this feat by cleverly using an inactive mutant that formed large, stable octameric complexes suitable for studying with cryo-em. Visualization of the transmembrane domain as well as the flexible periplasmic domain adds to our understanding of this class of co-polymerases and provides a solid basis for new ideas on how these structures control polysaccharide chain length.

This structure describes structural details for the first time for this sub-class of co-polymerases, is indeed beautiful, and the resolution achieved is impressive. However, the manuscript in its current form is difficult to follow and not accessible to anyone outside the LPS/CPS biosynthesis world. The manuscript seems to get a bit bogged down with too much details and unnecessary and often redundant or awkwardly organized figures. For example the authors get bogged down discussing the multiple sequence alignments - it is difficult to follow and most readers will be lost. It is not so important in the context of their new structure. Also For example fig.1a and fig.5c are basically the same. The space-filling representations in fig.2 do not bring any new information. Fig. 2e for example feels out of place in relation to the rest of that figure. Panel fig.3d would better fit in fig.2 (perhaps in place of 2e?). The main results (which are generally quite nice!) are getting drowned out with many of these things. I suggest to simplify and reorganize the manuscript to focus the new observations and new components to make it for accessible to a wider audience.

WT treated with Wzb does not look so bad in the negative stain images. Was a small data collection attempted and processed? I wonder if there is a small population of octamers that can be detected here? If only the K540M mutant makes octamers, might that not imply that the octameric form is inactive? If Wzc is somehow cycling between octamers and monomers or other oligomeric states you should at least be able to see a small amount of octamers even in the WT protein? The K540M elutes at about 13ml in the SEC compared to 15ml for wt. What does the side of the SEC peak at 13ml in wt look like under the microscope? Also what is the effect of the detergent on the formation of octamers? Where other detergents tried before settling on LMNG? Have you tried incorporating into nanodiscs?

Were any of the WT-YE mutations checked with negative staining or otherwise?

The authors discuss almost in passing the wzc transmembrane domain... it does indeed look very similar to that of WzzB suggesting a similar interaction with the polymerase wzy. But are there any differences that might suggest a slightly different interaction specific to wzc?

The flexible motif 3 is quite interesting. This flexibility seems to me due to the fact that it is missing wza that would otherwise stabilize it. Could pull down assays (or something similar) capture the periplasmic (or piece of) domain of wza?

Could it be possible to stabilize this motif? through mutations for example? The authors mention a gain of function RAG1 wzc equivalent R410L. What does this mutation do to motif 3? Class i and class ii conformations of motif3 appear to me as an open and closed state. Could R410L favor one of these conformations?

The authors attempt to understand the role of the tyrosine rich tail by constructing a series of tyrosine to glutamate mutations. They state that Glu would mimic a phosphotyrosine but do not explain why. The superposition of the 2 kinase domain looks misleading to me (fig.4e). As is, it appears that the big bulk of the protein is rotating as a result of the 'phosphotyrosines'. Isn't it more likely that the bulk of wzc is stationary and the kinase domain is rotating? (ie do a superposition using the TM domain

and/or periplasmic domain).

The authors over-reach in connecting the subtle movement of periplasmic domain to changes to the kinase domain as it is not very clear from the data presented here. Or they need to better explain how they believe they are connected. Indeed, if autophosphorylation is disassembling the octamer, the entire protein would be 'sensitive to changes in the kinase domain' not only motif3.

The authors suggest based on their 4YE structure that the tyrosine rich tail is essentially pulled through the active site phosphorylating each tyrosine as it does. And this pulling and phosphorylating is disrupting the octameric structure. If that is true it would stand to reason that in other YE mutants different tyrosine residues should be in the active site. I know it is not reasonable to ask for data collections on all of these, but could they get an idea from the negative stained images? Perhaps a percentage of octamers in each mutant? Or some other metric? As it stands the authors present 2 "extreme" structures with 2 different tyrosines in the active site. A third data point, (perhaps an intermediate structure?) would help sell this point.

Also, if the octamers are being destabilized in the 4YE mutant, can you see 7-mers or 6-mers? 5-mers? Etc. Figure 4c and extended data figure 9d suggests so. Can any of these be refined?

The proposed model of wzc opening wza to the periplasm and the "resulting conformational changes activating wzy" is highly speculative and seems quite vague... They do not state any evidence to support this other than 'integrating the available data'. What available data? I agree it is reasonable to assume that motif3 is interacting with wza and even potentially opening it. But how exactly? If the octameric complex engages with wza to open it, then simply disassembling the wzc octamer would cause wza to close - no 'communication' between the kinase domain and motif3 would be required in that case. Perhaps the structure of R410L would shed light on this or other interaction experiments with wza described above.

Reviewer #2:

Remarks to the Author:

In the manuscript of Yang an attempt is made to study the function and effect of C-terminal phosphorylation on the assembly of the bacterial capsule Wzc (PCP-2a) proteins. By looking with EM, gel filtration and mass spectrometry at the phosphorylation and assembly status of this E Coli protein they provide evidence that pTyr phosphorylation in the C-terminal tail hinders the octamerization of this protein. These findings are validated by Tyr mutants.

Basically, native mass spectrometry was used to measure the intact Wzc protein, to determine its phosphorylation status and "phosphoproteomics" was used to assess which sites were phosphorylated.

It is a nicely presented and easy to read study, although as outsider I have to say that the abstract is a bit hard to read if you are not an expert. Only later is explained what the Wzx-Wzy mechanism is for instance.

Here my comments on the MS data

The EM data is made available, but no reference is made to the availability of the proteomics/mass spectrometry data.

The data in Figure 1B do not look to convincing, at least a table with assigned masses and how much they deviate from the expected masses should be given. The phosphorylation stoichiometry seems to be the highest for the 5P, especially as no 6P is seen. How do the authors explain that?

Also, the annotation of the phosphopeptides should include localization scores, especially as the Tyr residues are so close to each other. The search was also only done with allowing pTyr. As pSer and pThr are way more abundant in E Coli that should at least be tested as well. The peptides shown in Extended Data Figure 2 show also Ser and His in this stretch. The authors should thus allow these modifications as well, as pHis is more prevalent in bacteria than pTyr. The parameter given in Ex Data Figure 2 reveal no evidence for size localization. Therefore, I would also argue that the underlying data should be deposited to be able to check the assignments. The spectrum in 2B may look OK, but without delta score or localization score given it is hard to judge.

In the extended Data Figure 8 negative-stain micrographs of WzcK540M with different Y(Tyr) to E (Glu) mutants are shown to reveal the oligomerization states. I do find this an odd way of assessing oligomerization. Why do they not use gel-filtration or size exclusion chromatography, or as they already do use it native mass spectrometry. Not only are the EM images hard to interpret, but they may also be less quantitative. It would be any way essential to complement the EM data with either gel filtration or native MS. I failed to read why the native MS of the octamer was not attempted or failed?

Reviewer #3:

Remarks to the Author:

Bacteria possess a unique class of membrane-spanning protein tyrosine kinases, the BY-kinases that work as part of a multiprotein machinery in the synthesis of high molecular-weight polysaccharides. In Gram-negative bacteria, the BY-kinases span the inner membrane and comprise of periplasmic, transmembrane and cytoplasmic components. While the structure of the cytoplasmic catalytic domain had been solved several years ago, no atomic resolution structure of a full-length BY-kinase was available, and this was a major gap in knowledge in the field. The authors, Yang et al. have now successfully filled this gap by providing a structure of the full-length BY-kinase Wzc. I feel that this represents a significant breakthrough in the field and should be of interest to the readers of Nature Communications. The manuscript is well written and well-illustrated. While I have no major concerns, I do have a few minor comments that will improve the manuscript if appropriately addressed:

1. A point that is somewhat unclear in the manuscript is that the authors seem to suggest (at least in my read) that phosphorylation is sequential. The authors should attempt to explain this point better. It is unclear to this reviewer how the authors can distinguish random from sequential events. Does the system gain any advantage structurally by following a certain sequence of phosphorylation events rather achieving the same overall level of phosphorylation leading to disassembly?
2. It is mentioned on page 8, that residues 65-84 of the periplasmic domain are missing from the EM structure, probably due to a highly mobility of this region. However, looking at the sequence alignment presented in the Extended Figure 5, it seems that this region seems to display a high level of conservation, especially in the 70-80 segment. Was the importance of this region for Wzc function i.e. polysaccharide synthesis or export or assembly tested using some mutation/deletion strategy?
3. It is worth listing RMSD values when comparing structural modules.
4. The precise definition of motifs in terms of residue numbers would be useful.
5. On page 3: the sentences – “Wzz (PCP-1) homologs are required for polymerization and regulation of chain length distributions of O-antigen polysaccharides. In contrast, Wzz (PCP-1) homologs are required for polymerization and regulation of chain length distributions of O-antigens in lipopolysaccharide glycolipids ...” seems repetitive.
6. Best to use the full name of the organism at its first occurrence e.g. on page 13 *Klebsiella pneumoniae* rather than *K. pneumoniae*.
7. On page 17: “to generate pBRC90” is in bold for some reason.
8. On page 18: “Lysates were prepared by harvesting 1 OD_{600nm}” is in italics for some reason.
9. On page 18, “Protran 0.45 im nitrocellulose membrane” should actually be “Protran 0.45 μm nitrocellulose membrane”

manuscript *Yang et al*;

We thank the reviewers for their careful reading of the manuscript and their helpful comments. Our response is based on supplying new data where ever possible.

Our point to point response to each review is in red:

REVIEWER 1

The authors present in this manuscript the near atomic resolution structure of an E.coli wzc polysaccharide co-polymerase involved in bacterial capsid assembly. There are a few structures of isolated kinase domains as well as a wzz structure describing the TM domain from this family of proteins; however this would be the first fully intact structure for this sub-class of PCP proteins. The authors accomplish this feat by cleverly using an inactive mutant that formed large, stable octameric complexes suitable for studying with cryo-em. Visualization of the transmembrane domain as well as the flexible periplasmic domain adds to our understanding of this class of co-polymerases and provides a solid basis for new ideas on how these structures control polysaccharide chain length.

This structure describes structural details for the first time for this sub-class of co-polymerases, is indeed beautiful, and the resolution achieved is impressive. However, the manuscript in its current form is difficult to follow and not accessible to anyone outside the LPS/CPS biosynthesis world. The manuscript seems to gets a bit bogged down with too much details and unnecessary and often redundant or awkwardly organized figures. For example the authors get bogged down discussing the multiple sequence alignments - it is difficult to follow and most readers will be lost. It is not so important in the context of their new structure. Also For example fig.1a and fig.5c are basically the same. The space-filling representations in fig.2 do not bring any new information. Fig. 2e for example feels out of place in relation to the rest of that figure. Panel fig.3d would better fit in fig.2 (perhaps in place of 2e?). The main results (which are generally quite nice!) are getting drowned out with many of these things. I suggest to simplify and reorganize the manuscript to focus the new observations and new components to make it for accessible to a wider audience.

We thank the reviewer for their comments. We have edited the manuscript extensively with the goal of improving the presentation for non-specialist, while still accommodating the large amount of new data. We reorganized the figures and now have only one space-filling image which is shown to emphasise the side portals. Discussion about the multiple sequence alignments has been reduced.

WT treated with Wzb does not look so bad in the negative stain images. Was a small data collection attempted and processed? I wonder if there is a small population of octamers that can be detected here? If only the K540M mutant makes octamers, might that not imply that the octameric form is inactive? If Wzc is somehow cycling between octamers and monomers or other oligomeric states you should at least be able to see a small amount of octamers even in the WT protein? The K540M elutes at about 13ml in the SEC compared to 15ml for wt. What does the side of the SEC peak at 13ml in wt look like under the microscope? Also what is the effect of the detergent on the formation of octamers? Where other detergents tried before settling on LMNG? Have you tried incorporating into nanodiscs?

This is a useful comment for which we are grateful and reply with data.

We have now collected a dataset of Wzb-treated Wzc (1787 micrographs under cryogenic condition using a Glacios EM; 356,511 particles are picked in total). We managed to classify as an octamer a very small portion (around 0.1%) of all particles. We have added this data to Extended Data Fig 1g. These data do show that dephosphorylation of native Wzc by its phosphatase Wzb results in octamer (though very small portion under these conditions). Our functional data show a clear correlation between the ability to form an octamer and polysaccharide production (Extended Data Fig 8).

We have looked material the SEC-peak side shoulder of WT Wzc using negative stain, and it has multiple different aggregate forms. We don't see any evidence for an octamer in the negative-stain micrographs.

We have tried nanodisc, but the sample starts to crash when adding the biobeads to remove the detergent. We reason it is due to diameter of the protein (the transmembrane region is ~100 Å in diameter, but with a large centre cavity) which is close to the maximum for most well understood MSPs (~150Å). We have tried DDM for Wzc^{K540M} but the protein although purified was difficult to get into the holes of grids. Since LMNG worked well, so no other detergents were evaluated.

The question of the “active form” is complicated. Biological data show that is the cycling between the phosphorylated and non-phosphorylated forms that IS required for function *in vivo*. Our data establish that the cycling of phosphorylation results in cycling between monomer and octamer. We favour the “active model” is where the octamer Wzc is present, since this is consistent with the octamer of Wza (it's partner) but this is a hypothesis that is now set up for future investigation. We have tried to make this clear in the presentation. The question asked is an interesting one but will likely require the structure of the Wza-Wzc complex to resolve and lies beyond the scope of this work.

Were any of the WT-YE mutations checked with negative staining or otherwise?

We checked Y to E mutants in native background for kinase activity and capsule assembly. These data are reported in the Extended Data Fig 8b-e. We tested these mutant proteins in mass spectrometry, where we saw the monomer (shown in Extended Data Fig 2c) in the same way as native Wzc. Since the Y to E mutants in native background behaved as native protein in mass spectrometry and functionally, we did not pursue any further analysis.

All the Y to E mutants for the Wzc^{K540M} background were checked in negative staining.

Our key point is that when enough Y to E changes are introduced, the octamer seen in the dephosphorylated form is unstable and the function is lost.

The authors discuss almost in passing the wzc transmembrane domain... it does indeed look very similar to that of WzzB suggesting a similar interaction with the polymerase wzy. But are there any differences that might suggest a slightly different interaction specific to wzc?

We have expanded the discussion on comparison to Wzz. We think that interpreting structural differences in terms of altered interactions with the polymerase in the two systems would be premature as this needs a lot more structural, functional and analytical data.

The flexible motif 3 is quite interesting. This flexibility seems to me due to the fact that it is missing wza that would otherwise stabilize it. Could pull down assays (or something similar) capture the periplasmic (or piece of) domain of wza?
Could it be possible to stabilize this motif? through mutations for example?

We agree the motif 3 is very interesting, and it is very likely the key for Wzc-Wza coordination. We have shown it is critical for overall function and we have shown that this can be uncoupled from the kinase activity (Fig. 5b). This sets the stage for further studies to examine the interactions but there is a lot of work to do to identify how Wza interacts with Wzc. Our model and existing data suggests that the complex may be transitory requiring particular phosphorylation state and /or Wzy. We assume the formation of the Wzc octamer with Wzy is important to engage Wza. We respectfully suggest such studies lie beyond the scope of the current investigation.

The authors mention a gain of function RAG1 wzc equivalent R410L. What does this mutation do to motif 3? Class i and class ii conformations of motif3 appear to me as an open and closed state. Could R410L favor one of these conformations?

We have made the R410L mutant but did not see any difference using gel analysis of the polymer product. This is hard to interpret because of differences in the sequences of the two homologs. Given this lack of a phenotype (to link structure and biology in a specific mutant), we believe pursuing another full EM structure determination is not justified

The authors attempt to understand the role of the tyrosine rich tail by constructing a series of tyrosine to glutamate mutations. They state that Glu would mimic a phosphotyrosine but do not explain why. The superposition of the 2 kinase domain looks misleading to me (fig.4e). As is, it appears that the big bulk of the protein is rotating as a result of the 'phosphotyrosines'. Isn't it more likely that the bulk of wzc is stationary and the kinase domain is rotating? (ie do a superposition using the TM domain and/or periplasmic domain).

We apologise for the lack of clarity. Our goal was to create a situation resembling a locked phosphorylated state with the premise that the Glu mutation mimics (imperfectly) the negatively charged phosphotyrosine. This is now more clearly stated. Since Wzc system auto-phosphorylates, mutation is the only way to produce defined distinct states. The signal resulting from phosphorylation states in the kinase domain is transduced through the TM to the periplasmic region. We believe that the superposition of the kinase domain informs these insights. We have clarified what the super-positions are intended to show and have added quantitative measures.

The authors over-reach in connecting the subtle movement of perplasmic domain to changes to the kinase domain as it is not very clear from the data presented here. Or they need to better explain how they believe they are connected. Indeed, if autophosphorylation is disassembling

the octamer, the entire protein would be ‘sensitive to changes in the kinase domain’ not only motif3.

We have reworded the relevant text and apologise for the lack of clarity. It is of course true that the dissociation of the octamer changes the entire structure. However, what we were trying to convey was there were indeed rigid body shifts between Wzc^{K540M} and $Wzc^{K540M}4YE$. These shifts arise from changes in the kinase domain. We have now added more quantitative description to make this point.

The authors suggest based on their 4YE structure that the tyrosine rich tail is essentially pulled through the active site phosphorylating each tyrosine as it does. And this pulling and phosphorylating is disrupting the octameric structure. If that is true it would stand to reason that in other YE mutants different tyrosine residues should be in the active site. I know it is not reasonable to ask for data collections on all of these, but could they get an idea from the negative stained images? Perhaps a percentage of octamers in each mutant? Or some other metric? As it stands the authors present 2 “extreme” structures with 2 different tyrosines in the active site. A third data point, (perhaps an intermediate structure?) would help sell this point.

We accept that we should have have presented our argument more clearly and thank the referee for their suggestion.

The mass spectrometry data showed that all conserved tyrosines can be phosphorylated and therefore thus MUST at some point enter the active site. In this study, these two “extreme” structures were used (non-phosphorylated with Y717 at the active site, and multiply phosphorylated with Y708 at the active site) to gain insight how the phosphorylation could proceed (Y717, Y715, Y713, Y708). The previously determined crystal structure showed Y715 tyrosine at the active site, and we have discussed the structural changes that result from the shift Y717 to Y715 at the active site. There are three data points.

The only structure “missing” is Y713 at the active site. Since negative stain is unable to resolve such fine detail, filling this gap would require another full structural determination. Given the point we are making and the data in hand, we argue the additional full structural determination would add limited further insight while representing a substantial effort.

We have calculated the percentage of octamers in each mutant in the Extended Data Fig 8a, by analysing the negative-stain micrographs.

We believe that the phosphorylation most commonly proceeds from C-terminal to N-terminal (from Y717, then Y715, then Y713 and finally Y708). This is because when Y715 is phosphorylated (pY715) at the active site, it has to move to allow ATP to enter the active site. If it moves to allow unmodified Y717 to enter the active site, then pY715 would be located at a small pocket which has a negative charge (the pocket is described in the Wzc^{K540M} structure, new Figure 2g). This is unfavourable. A similar analysis holds when pY713 is at the active site, to allow Y715 access to the active site pY713 would have to move into this pocket and set up unfavourable interactions. Thus, we argue for a process involving pY717 first, followed by pY715, and then by pY713.

Also, if the octamers are being destabilized in the 4YE mutant, can you see 7-mers or 6-mers? 5-mers? Etc. Figure 4c and extended data figure 9d suggests so. Can any of these be refined?

The 2D averages do suggest a high level of conformational heterogeneity and flexibility. We have looked at this again but we were unable to identify uniform particles with the data we have in hand. We cannot definitively exclude the possibility of other oligomers, but we have no evidence for them.

The proposed model of wzc opening wza to the periplasm and the “resulting conformational changes activating wzy” is highly speculative and seems quite vague... They do not state any evidence to support this other than ‘integrating the available data’. What available data? I agree it is reasonable to assume that motif3 is interacting with wza and even potentially opening it. But how exactly? If the octameric complex engages with wza to open it, then simply disassembling the wzc octamer would cause wza to close - no 'communication' between the kinase domain and motif3 would be required in that case. Perhaps the structure of R410L would shed light on this or other interaction experiments with wza described above.

We largely agree with the referee’s proposed model which is more elegantly expressed than we managed in the first version. We were trying to present a hypothesis to highlight directions for future investigation, rather a firm conclusion. We have reduced the speculative discussion to address the concerns.

REVIEWER 2

In the manuscript of Yang an attempt is made to study the function and effect of C-terminal phosphorylation on the assembly of the bacterial capsule Wzc (PCP-2a) proteins. By looking with EM, gel filtration and mass spectrometry at the phosphorylation and assembly status of this E Coli protein they provide evidence that pTyr phosphorylation in the C-terminal tail hinders the octamerization of this protein. These findings are validated by Tyr mutants.

Basically, native mass spectrometry was used to measure the intact Wzc protein, to determine its phosphorylation status and “phosphoproteomics” was used to assess which sites were phosphorylated.

We thank the reviewer for their helpful and preceptive comments.

It is a nicely presented and easy to read study, although as outsider I have to say that the abstract is a bit hard to read if you are not an expert. Only later is explained what the Wzx-Wzy mechanism is for instance.

The abstract has been reworded.

The EM data is made available, but no reference is made to the availability of the proteomics/mass spectrometry data.

The mass spectrometry proteomics data have been deposited to the ProteomeXchange Consortium via the PRIDE partner repository with the dataset identifier PXD025820 and here are the reviewer account details (Username: reviewer_pxd025820@ebi.ac.uk, Password: aXS6XUHa). The pointer is now in the text under the data availability section. We apologise for not communicating this earlier, the referee is quite right about best practice.

The data in Figure 1B do not look to convincing, at least a table with assigned masses and how much they deviate from the expected masses should be given. The phosphorylation stoichiometry seems to be the highest for the 5P, especially as no 6P is seen. How do the authors explain that?

A valid point and we have now added a table with the expected and the measured masses in Fig 1b. There are are potential additional sites (other than 4 conserved tyrosines) available for phosphorylation but interestingly both in our native MS and proteomics data we have seen only a maximum of 5 sites getting phosphorylated consistently. The sample we used is Wzc WT in detergent.

The very low / no abundance of 6P species (both in mass spec and in the 5YE mutant in Figure 4b) is what is expected from the “tipping point” model. The complex falls apart, no further modification of tyrosine tail.

Also, the annotation of the phosphopeptides should include localization scores, especially as the Tyr residues are so close to each other. The search was also only done with allowing pTyr.

As pSer and pThr are way more abundant in E Coli that should at least be tested as well. The peptides shown in Extended Data Figure 2 show also Ser and His in this stretch. The authors should thus allow these modifications as well, as pHis is more prevalent in bacteria than pTyr. The parameter given in Ex Data Figure 2 reveal no evidence for size localization. Therefore, I would also argue that the underlying data should be deposited to be able to check the assignments. The spectrum in 2B may look OK, but without delta score or localization score given it is hard to judge.

We thank the reviewer for this suggestion. Phospho site probabilities from Maxqaunt software are now included in the table (extended data figure 2a). As suggested, we performed additional search with allowing pSer, pThr, pTyr and pHis. Although the software assigned phosphorylation to Ser and His within the peptide 708-718 given their close proximity this could be an error and would be inconsistent with extant literature on antibody western blots.

To further validate this assignment, in addition to checking the data manually, we ran phosphoproteomics on several mutants (Y717E-Y718E, Y715E-Y717E-Y718E, Y713E-Y715E-Y717E-Y718E, and Y708E-Y713E-Y715E-Y717E-Y718E) where the conserved tyrosine residues are mutated sequentially to glutamic acids and the data suggest that the number of phosphosites decreases as the number of mutations increase (a similar result was also observed by native MS shown in extended data figure 2c). Interestingly, pSTYH search did not assign the phosphorylation on Ser and His in the samples where the tyrosine residues next to/nearby Ser/His were mutated to glutamic acid (shown below RF1). We therefore concluded that the phosphorylation by Wzc is specific to Tyr only. We have added the following statement in the methods section to clarify our phosphosite search with pTyr and deposited all these mutants' data on PRIDE.

The revision now states

“our initial analysis with pSYTH search yielded the assignment of phosphosites on His and Ser along with conserved Tyr residues within the peptide 708-718 given their close proximity, however, based on manual inspection and phosphoproteomics analyses on several mutants (data is on PRIDE with accession number PXD025820) suggested that the assignment on His and Ser was not correct. Thus, we used pY only for searching phosphosites in our subsequent analyses”.

RF1 MS/MS spectra of the phosphopeptide (708-718). From a) wildtype, b) Y717E-Y718E, c) Y715E-Y717E-Y718E, d) Y708E-Y713E-Y715E-Y717E-Y718E. Search was performed with pSYTH and phosphosites were assigned on His and Ser in case of wild type but not in the cases where nearby tyrosines were mutated.

In the extended Data Figure 8 negative-stain micrographs of WzcK540M with different Y(Tyr) to E (Glu) mutants are shown to reveal the oligomerization states. I do find this an odd way of assessing oligomerization. Why do they not use gel-filtration or size exclusion chromatography, or as they already do use it native mass spectrometry. Not only are the EM images hard to interpret, but they may also be less quantitative. It would be any way essential to complement the EM data with either gel filtration or native MS. I failed to read why the native MS of the octamer was not attempted or failed?

We have provided some representative gel filtration profiles of different mutants (see figure RF2 below). Gel filtration is not very informative (see native and K540M in Extended Data Fig 1) for this protein; there is a relatively small shift in the retention time (native protein is smaller and elutes later but the difference is not that of an octamer to monomer). Gel filtration reports on the detergent micelle as well as the protein. This becomes more complicated when the samples contain different oligomerization states (which is the case for Wzc^{K540M} Y to E mutants).

We know from EM there are ill defined aggregates in the native protein. This level of insight is absent from gel filtration. We have modified the EM grid figures by introducing circles indicating the representative octamers.

The quantitation is a good question and to address that, we analysed around 10 negative-stain micrographs for each mutant (a significant undertaking) to calculate ratio of

octamers to the number of particles (Extended Data Fig 8a). The octamer is judged by the size and shape of the particle. This is a crude estimate and could be confounded by strange orientation effects etc; however, it does show a trend consistent with the thrust of the paper.

In Extended Data Fig 8a (negative-stain), it represents the ratio of octamers to total particles (which would contain both proteins and free detergent micelles) on negative-stain micrographs. In Extended Data Fig 9 (cryo EM), it represents the ratio of octamers inside all classified particles (a more accurate approach). These measures are different. A full cryo EM analysis of all mutants we do not feel would add to the paper.

RF2 Gel filtration profiles of representative Y to E mutants, including WT (black), Wzc^{K540M} (red), and Single (Y715E, cyan) or multiple (3YE, green; 4YE, blue; 5YE, salmon) Y to E mutants on Wzc^{K540M}.

We have attempted native MS of the K540M in several detergents (C8E4, LDAO and G1) that usually require less amount of energy to remove the detergent micelles (so called MS friendly). However, Wzc K540M remained tightly associated with LMNG. In all cases we obtained a broad hump peak. We were unable to resolve the hump even after using all the energies available on various Q-Exactive instruments as it proved impossible to remove LMNG from the protein (hence the micelle hump). This behaviour is strikingly different than the wild type protein (also shown for comparison in RF3). We therefore regarded this as inconclusive (but consistent with an octamer) and did not report it in the main manuscript.

RF3 Native MS analyses of Wzc-K540M in various detergents (a-c) and wild type (d). The hump in the region $m/z=12000-16000$ could not be resolved due to lack of sufficient energies to remove the LMNG micelles incorporated during purification. The higher molecular weight is consistent with an octamer but we do not regard the data as definitive and did not include in the main manuscript.

REVIEWER 3

Bacteria possess a unique class of membrane-spanning protein tyrosine kinases, the BY-kinases that work as part of a multiprotein machinery in the synthesis of high molecular-weight polysaccharides. In Gram-negative bacteria, the BY-kinases span the inner membrane and comprise of periplasmic, transmembrane and cytoplasmic components. While the structure of the cytoplasmic catalytic domain had been solved several years ago, no atomic resolution structure of a full-length BY-kinase was available, and this was a major gap in knowledge in the field. The authors, Yang et al. have now successfully filled this gap by providing a structure of the full-length BY-kinase Wzc. I feel that this represents a significant breakthrough in the field and should be of interest to the readers of Nature Communications. The manuscript is well written and well-illustrated. While I have no major concerns, I do have a few minor comments that will improve the manuscript if appropriately addressed:

We thank the reviewer for their supportive comments and address their criticisms below.

1. A point that is somewhat unclear in the manuscript is that the authors seem to suggest (at least in my read) that phosphorylation is sequential. The authors should attempt to explain this point better. It is unclear to this reviewer how the authors can distinguish random from sequential events. Does the system gain any advantage structurally by following a certain sequence of phosphorylation events rather achieving the same overall level of phosphorylation leading to disassembly?

Sequential phosphorylation is plausible since one of the pockets prior to the active site would seem incompatible with binding of phosphorylated residue. However, the referee is quite correct, we have not “proven” it. We have modified the discussion.

2. It is mentioned on page 8, that residues 65-84 of the periplasmic domain are missing from the EM structure, probably due to a highly mobility of this region. However, looking at the sequence alignment presented in the Extended Figure 5, it seems that this region seems to display a high level of conservation, especially in the 70-80 segment. Was the importance of this region for Wzc function i.e. polysaccharide synthesis or export or assembly tested using some mutation/deletion strategy?

We have not yet mutated these residues. We agree it is likely important and we think the target for future studies.

3. It is worth listing RMSD values when comparing structural modules.

We have done so and think it add clarity, a very useful suggestion.

4. The precise definition of motifs in terms of residue numbers would be useful.

This is now explicitly stated.

5. On page 3: the sentences – “Wzz (PCP-1) homologs are required for polymerization and regulation of chain length distributions of O-antigen polysaccharides. In contrast, Wzz (PCP-

1) homologs are required for polymerization and regulation of chain length distributions of O-antigens in lipopolysaccharide glycolipids ...” seems repetitive.

Modified

6. Best to use the full name of the organism at its first occurrence e.g. on page 13 *Klebsiella pneumoniae* rather than *K. pneumoniae*.

Modified

7. On page 17: “to generate pBRC90” is in bold for some reason.

This seems to be a strange conversion error; the word file is normal.

8. On page 18: “Lysates were prepared by harvesting 1 OD_{600nm}” is in italics for some reason.

This seems to be a strange conversion error; the word file is normal.

9. On page 18, “Protran 0.45 im nitrocellulose membrane” should actually be “Protran 0.45 μm nitrocellulose membrane”

This seems to be a strange conversion error; the word file is normal

Reviewers' Comments:

Reviewer #1:

Remarks to the Author:

The manuscript has been substantially improved, focuses on the new data and is to the point. These improvements have increased its readability and accessibility to a wider audience. My comments have been adequately addressed.

A few things I noticed:

Lines 52-54. "Although the structures..." this sentence is awkward... missing a word?

Extended figure 1g: The zoomed octameric particle is hard to notice. At first glance it looks like junk on the micrograph. Maybe put a border around it to make it stand out better, or move it to the outside of the micrograph.

Line 131: "made structure determination was impractical" -remove "was"?

Lines: 220-222, Figure 3c / Extended figure 6d: comparing the R410-R262 Wzc and R279-R98 Wzz. Maybe put the associated figure panels side-by-side in the same figure for easier comparison. As is, in separate figures, it is a bit cumbersome. Also what PDB was used for Wzz? Perhaps include in figure legend and/or text.

Figure 2f/line 623: "key residues are labelled in italics" -however in the figure panel I cannot distinguish the italics from non-italics residues.

Reviewer #2:

Remarks to the Author:

In response to my review the authors have uploaded their data to Pride and performed searches allowing also pSer, pThr and pHis. They noticed that this led to assignments different from the pTyr they assigned. In general this shows how difficult it is to assign phosphorylation sites in multiple phosphorylated peptides containing a lot of potential phosphate acceptor sites. They argue that their assignments were validated by manual inspection and by site-directed mutations. The first argument I do not support, but the mutational data gives indeed credence to their assignments. So in general I am happy with their additional work supporting their assignments. Also the additions made for the native MS analysis, although not successful in answering the oligomerization status is well appreciated.

manuscript *Yang et al*;

We thank the two reviewers for their careful reading of the revised manuscript.

Our point to point response to each review is in red:

REVIEWER 1

REVIEWERS' COMMENTS

Reviewer #1 (Remarks to the Author):

The manuscript has been substantially improved, focuses on the new data and is to the point. These improvements have increased its and accessibility to a wider audience. My comments have been adequately addressed.

We thank the reviewer for their comments and their input has improved the manuscript.

A few things I noticed:

Lines 52-54. "Although the structures..." this sentence is awkward... missing a word?

Inserted the missing word "by".

Extended figure 1g: The zoomed octameric particle is hard to notice. At first glance it looks like junk on the micrograph. Maybe put a border around it to make it stand out better, or move it to the outside of the micrograph.

Boxed in Figure ED1.

Line 131: "made structure determination was impractical" -remove "was"?

Removed

Lines: 220-222, Figure 3c / Extended figure 6d: comparing the R410-R262 Wzc and R279-R98 Wzz. Maybe put the associated figure panels side-by-side in the same figure for easier comparison. As is, in separate figures, it is a bit cumbersome.

Panel now shifted to Figure 3d

Also what PDB was used for Wzz? Perhaps include in figure legend and/or text.

PDB 6RBG, now stated

Figure 2f/line 623: "key residues are labelled in italics" -however in the figure panel I cannot distinguish the italics from non-italics residues.

The legend has been modified and the residues in bold and italics.

Reviewer #2 (Remarks to the Author):

In response to my review the authors have uploaded their data to Pride and performed searches allowing also pSer, pThr and pHis. They noticed that this lead to assignments different from the pTyr they assigned. In general this shows how difficult it is to assign phosphorylation sites in multiple phosphorylated peptides containing a lot of potential phosphate acceptor sites.

They argue that their assignments were validated by manual inspection and by site-directed mutations. The first argument I do not support, but the mutational data gives indeed credence to their assignments. So in general I am happy with their additional work supporting their assignments.

Also the additions made for the native MS analysis, although not successful in answering the oligomerization status is well appreciated

We thank the reviewer for their comments and their input has improved the manuscript.

We make no specific change in the manuscript in response.